# Novel EDGE encoding method enhances ability to identify genetic interactions

Molly A. Hall[1,2,3]*, John Wallace[1], Anastasia M. Lucas[4], Yuki Bradford[4], Shefali S. Verma[4], Bertram Müller-Myhsok[5,6,7], Kristin Passero[2], Jiayan Zhou[1], John McGuigan[1], Beibei Jiang[5,6,7], Sarah A. Pendergrass[8], Yanfei Zhang[9], Peggy Peissig[10], Murray Brilliant[10], Patrick Sleiman[11], Hakon Hakonarson[11], John B. Harley[12,13,14], Krzysztof Kiryluk[15], Kristel Van Steen[16,17‡], Jason H. Moore[4‡], Marylyn D. Ritchie[4‡]

1 Department of Veterinary and Biomedical Sciences, College of Agricultural Sciences, The Pennsylvania State University, University Park, Pennsylvania, United States of America, 2 Huck Institutes of the Life Sciences, The Pennsylvania State University, University Park, Pennsylvania, United States of America, 3 Penn State Cancer Institute, The Pennsylvania State University, University Park, Pennsylvania, United States of America, 4 Department of Genetics, Institute for Biomedical Informatics, University of Pennsylvania, Philadelphia, Pennsylvania, United States of America, 5 Department of Translational Research in Psychiatry, Max Planck Institute of Psychiatry, Munich, Germany, 6 Munich Cluster for Systems Neurology (SyNergy), Munich, Germany, 7 Institute of Translational Medicine, University of Liverpool, Liverpool, United Kingdom, 8 Genentech, San Francisco, California, United States of America, 9 Genomic Medicine Institute, Geisinger Health System, Danville, Pennsylvania, United States of America, 10 Center for Precision Medicine Research, Marshfield Clinic Research Institute, Marshfield, Wisconsin, United States of America, 11 Department of Pediatrics, Center for Applied Genomics, Children's Hospital of Pennsylvania, Philadelphia, Pennsylvania, United States of America, 12 Center for Autoimmune Genomics and Etiology (CAGE), Cincinnati Children's Hospital Medical Center, Cincinnati, Ohio, United States of America, 13 Department of Pediatrics, University of Cincinnati College of Medicine, Cincinnati, Ohio, United States of America, 14 United States Department of Veterans Affairs Medical Center, Cincinnati, Ohio, United States of America, 15 Division of Nephrology, Department of Medicine, College of Physicians and Surgeons, Columbia University, New York, New York, United States of America, 16 WELBIO, GIGA-R Medical Genomics-BIO3, University of Liège, Liège, Belgium, 17 Department of Human Genetics, University of Leuven, Leuven, Belgium

‡ These authors are joint senior authors on this work.
* mah546@psu.edu

## Abstract

Assumptions are made about the genetic model of single nucleotide polymorphisms (SNPs) when choosing a traditional genetic encoding: additive, dominant, and recessive. Furthermore, SNPs across the genome are unlikely to demonstrate identical genetic models. However, running SNP-SNP interaction analyses with every combination of encodings raises the multiple testing burden. Here, we present a novel and flexible encoding for genetic interactions, the elastic data-driven genetic encoding (EDGE), in which SNPs are assigned a heterozygous value based on the genetic model they demonstrate in a dataset prior to interaction testing. We assessed the power of EDGE to detect genetic interactions using 29 combinations of simulated genetic models and found it outperformed the traditional encoding methods across 10%, 30%, and 50% minor allele frequencies (MAFs). Further, EDGE maintained a low false-positive rate, while additive and dominant encodings demonstrated inflation. We evaluated EDGE and the traditional encodings with genetic data from the Electronic Medical Records and Genomics (eMERGE) Network for five phenotypes: age-related macular degeneration (AMD), age-related cataract, glaucoma, type 2 diabetes

**Data Availability Statement:** The result files for the eMERGE data are within the manuscript and its Supporting Information files. Scripts to reproduce the simulated data are available at https://github.com/HallLab/PLOS_Genetics_EDGE_Paper. All

data underlying paper's figures are available in Supporting Information files (eMERGE results) and https://github.com/HallLab/PLOS_Genetics_EDGE_Paper (simulation results). All eMERGE genotype and phenotype files are available from the dbGAP database https://www.ncbi.nlm.nih.gov/projects/gap/cgi-bin/study.cgi?study_id=phs000888.v1.p1 Scripts to reproduce the simulated data are available at https://www.hall-lab.org/. (Please note access to the dbGAP database is restricted).

**Funding:** The project described was partially supported by NIH grants LM010098 and AI116794 to JHM. KVS acknowledges funding from the Fonds de la Recherche Scientifique (FNRS) and opportunities offered by the interuniversity research institute WELBIO. The eMERGE Network was initiated and funded by NHGRI through the following grants: U01HG006828 (Cincinnati Children's Hospital Medical Center/Boston Children's Hospital) to JBH; U01HG006830 (Children's Hospital of Philadelphia) to HH; U01HG006389 (Essentia Institute of Rural Health, Marshfield Clinic Research Foundation and Pennsylvania State University); U01HG006382 (Geisinger Clinic); U01HG006375 (Group Health Cooperative/University of Washington); U01HG006379 (Mayo Clinic); U01HG006380 (Icahn School of Medicine at Mount Sinai); U01HG006388 (Northwestern University); U01HG006378 (Vanderbilt University Medical Center); and U01HG006385 (Vanderbilt University Medical Center serving as the Coordinating Center); U01HG004438 (CIDR) and U01HG004424 (the Broad Institute) serving as Genotyping Centers; and the PGRNSeq dataset (eMERGE PGx) for data collection. This work was additionally supported by the USDA National Institute of Food and Agriculture and Hatch Appropriations under Project #PEN04275 and Accession #1018544, startup funds from the College of Agricultural Sciences, Pennsylvania State University (https://agsci.psu.edu/), and the Dr. Frances Keesler Graham Early Career Professorship from the Social Science Research Institute, Pennsylvania State University (https://ssri.psu.edu/) to MAH. The funders had no role in study design, data collection and analysis, decision to publish, or preparation of the manuscript.

**Competing interests:** I have read the journal's policy and the authors of this manuscript have the following competing interests: MDR is on the scientific advisory board for Cipherome and Goldfinch Bio. The other co-authors have declared that no competing interests exist.

(T2D), and resistant hypertension. A multi-encoding genome-wide association study (GWAS) for each phenotype was performed using the traditional encodings, and the top results of the multi-encoding GWAS were considered for SNP-SNP interaction using the traditional encodings and EDGE. EDGE identified a novel SNP-SNP interaction for age-related cataract that no other method identified: rs7787286 (MAF: 0.041; intergenic region of chromosome 7)–rs4695885 (MAF: 0.34; intergenic region of chromosome 4) with a Bonferroni LRT p of 0.018. A SNP-SNP interaction was found in data from the UK Biobank within 25 kb of these SNPs using the recessive encoding: rs60374751 (MAF: 0.030) and rs6843594 (MAF: 0.34) (Bonferroni LRT p: 0.026). We recommend using EDGE to flexibly detect interactions between SNPs exhibiting diverse action.

## Author summary

Although traditional genetic encodings are widely implemented in genetics research, including in genome-wide association studies (GWAS) and epistasis, each method makes assumptions that may not reflect the underlying etiology. Here, we introduce a novel encoding method that estimates and assigns an individualized data-driven encoding for each single nucleotide polymorphism (SNP): the elastic data-driven genetic encoding (EDGE). With simulations, we demonstrate that this novel method is more accurate and robust than traditional encoding methods in estimating heterozygous genotype values, reducing the type I error, and detecting SNP-SNP interactions. We further applied the traditional encodings and EDGE to biomedical data from the Electronic Medical Records and Genomics (eMERGE) Network for five phenotypes, and EDGE identified a novel interaction for age-related cataract not detected by traditional methods, which replicated in data from the UK Biobank. EDGE provides an alternative approach to understanding and modeling diverse SNP models and is recommended for studying complex genetics in common human phenotypes.

## Introduction

Choosing between traditional methods for encoding single nucleotide polymorphisms (SNPs) in association studies, including additive, dominant, and recessive, requires making an assumption about the manner in which the coded risk allele acts. In accordance with Mendel's patterns of inheritance [1], given referent allele, $A$, and alternate (or coded risk) allele, $a$, all encodings assume that the $AA$ (homozygous referent) genotype incurs no risk and $aa$ (homozygous alternate) genotype bears full risk. As has been described previously [2–4], the assumed heterozygous ($Aa$) risk, however, varies according to the chosen encoding method. For each encoding, the assumed risk accrued by one copy of the alternate allele ($Aa$) in relation to two copies ($aa$) varies: $Aa$ is coded to bear 0%, 50%, or 100% the risk of $aa$ for recessive, additive, and dominant encodings, respectively. Codominant encoding is a dummy encoding method which allows $Aa$ and/or $aa$ to bear full risk. However, heterozygous risk could, in actuality, lie anywhere between 0% to 100% of the risk of a homozygous alternate genotype (for some examples of possible underlying genetic models, see Table 1). Additionally, choosing only one of these encodings is restrictive, as SNPs across the genome are unlikely to demonstrate identical genetic models. Testing all encodings increases the computational and multiple testing burden, thereby limiting the ability to identify true signals. This issue becomes more complicated

**Table 1. Examples of possible proportional genotype risk underlying genetic loci.**

| Biological Action | Homozygous Referent $AA$ | Heterozygous $Aa$ | Homozygous Alternate $aa$ |
|---|---|---|---|
| Recessive (REC) | 0% | 0% | 100% |
| Sub-Additive (SUB) | 0% | 25% | 100% |
| Additive (ADD) | 0% | 50% | 100% |
| Super-Additive (SUP) | 0% | 75% | 100% |
| Dominant (DOM) | 0% | 100% | 100% |

when dealing with epistasis (genetic interactions): testing all combinations of encodings for every SNP in a SNP-SNP interaction pair raises the multiple test burden and false negative rate. Here, we grapple with a limitation to studying epistasis: that genetic association tests are performed using genotype encoding methods that rely on assumptions about genetics that may not be reflected in the data, and thus, many genetic interactions likely remain elusive [4].

Since the advent of genome-wide association studies (GWAS) in 2005 [5], the human genetics community has widely adopted SNPs as a popular marker for genetic association studies [2]. Early GWAS involved comparisons of allele frequencies between cases and controls [5–7], and in 2006 the first GWAS to employ genotype encoding was published by Arking *et al.*, [8] using the additive, dominant, and recessive encodings. Following these studies many GWAS were performed using at least two of these three encodings [9–15], and in 2007, Bierut *et al.* published the first GWAS to use the additive encoding alone [16]. By 2008, use of the additive encoding alone was becoming commonplace for GWAS [2,3,17,18] with some researchers in the field arguing for methods that explore nonadditive encodings since [2,3,19]. The epistasis research community has largely adopted an additive encoding framework as well over the years. While some early studies of SNP-SNP interactions, assessed multiple genetic models [20–24], the shift to additive-only SNP encoding for regression-based epistasis occurred in 2008 [25–27]. Given the focus on the additive encoding for GWAS and epistasis research over the last decade, the work presented in this paper aims to 1) present a novel encoding that is flexible to detect SNPs with a nonadditive allelic architecture, 2) find evidence of SNPs that may act beyond an additive genetic model, 3) evaluate the different genetic encodings in the context of epistasis, and 4) identify novel SNP-SNP interactions associated with complex disease.

In this study, we introduce an alternative to the current paradigm: the elastic data-driven genetic encoding (EDGE), a method to flexibly assign each SNP with a unique heterozygous encoding prior to interaction analysis. We compared power and type I error of the current paradigm methodologies to EDGE with a simulation study. EDGE and the traditional encodings were applied to five disease phenotypes in the Electronic Medical Record and Genomics (eMERGE) Network [28,29]: age-related macular degeneration (AMD), age-related cataract, glaucoma, type 2 diabetes (T2D), and resistant hypertension and significant SNP-SNP interaction models were considered for replication in data from the UK Biobank [30]. Results demonstrate the benefit of investigating epistasis with methods beyond the additive encoding. We offer EDGE as a novel, flexible encoding method with the potential to identify elusive genetic interactions underlying complex diseases.

## Results

### EDGE accurately encodes SNPs according to simulated genetic models

Like traditional encodings, EDGE encodes the homozygous referent and homozygous alternate genotypes as 0 and 1, respectively. For additive, dominant, and recessive encodings, a

**Table 2. Recessive, additive, dominant, codominant and EDGE encoding schemes, scaled between 0 and 1.**

| Encoding | | Homozygous Referent *AA* | Heterozygous *Aa* | Homozygous Alternate *aa* |
|---|---|---|---|---|
| Traditional | Recessive | 0 | 0 | 1 |
| | Additive | 0 | 0.50 | 1 |
| | Dominant | 0 | 1 | 1 |
| | Codominant | 0 | 1 | 0 |
| | | 0 | 0 | 1 |
| EDGE | | 0 | α | 1 |

predetermined heterozygous value is assigned (Table 2). In contrast, EDGE assigns a flexible heterozygous value (α) based on the genetic model each individual SNP demonstrates in the dataset (Table 2). For example, if *SNP A*'s underlying genetic model is additive, the heterozygous genotype bears 50% the risk of homozygous alternate genotype (Table 1), and EDGE will assign the heterozygous genotype as 0.5. If *SNP B* demonstrates a sub-additive underlying model, the heterozygous genotype incurs 25% the risk of the homozygous alternate genotype, and the EDGE-derived alpha will be assigned as 0.25. In the case of *SNP B*, the heterozygous risk does not fall within the traditional encoding methods that are listed in Table 2. EDGE also can detect cases in which *SNP C*'s heterozygous genotype bears more risk than the homozygous alternate genotype, where the underlying model is over-dominant (α > 1) or in which *SNP D*'s heterozygous genotype bears less risk than the homozygous referent genotype (under-recessive; α < 0). Encoding detection and assignment occurs prior to interaction analysis, allowing for an encoding that reflects the underlying model of each unique SNP. EDGE is available for download in PLATO software [31] on the Ritchie Lab website https://ritchielab.org/plato.

The mechanisms of the EDGE method are described in the following steps:

1. Logistic or linear regression is run using a codominant (dummy) encoding with no intercept (so mean-centered trait; Y-E [Y] or prob (Y = 1)–pop prevalence). Note: if covariates are included in the analysis for adjustment, they are also included in the regression model and that, for consistency, we code the minor allele as the risk (alternate) allele.

$$Y = \beta_{Het} SNP_{Het} + \beta_{HA} SNP_{HA} \tag{1}$$

2. Using the beta values from the heterozygous genotype (Het) and homozygous alternate genotype (HA) dummy encodings, a weighted value (α) for the heterozygous genotype is calculated, whereby the α corresponds to the risk of the heterozygous genotype relative to homozygous alternate genotype when homozygous alternate risk is scaled to 1.

$$\alpha = \beta_{Het} / \beta_{HA} \tag{2}$$

3. These EDGE genotype encodings (homozygous referent = 0, heterozygous = α, homozygous alternate = 1) are then used for SNP-SNP interaction analysis. A common approach for genetic interaction is performing a likelihood ratio test (LRT) between a full and reduced model: Y = β0 + β1SNP1 + β2SNP2 + β3SNP1×SNP2 (full) and Y = β0 + β1SNP1 + β2SNP2 (reduced). A significant LRT p-value indicates that including the interaction term (β3 of the full model) significantly improves model fit when compared to a model

containing only the main effects of the two SNPs ($\beta_1$ and $\beta_2$). It has been previously shown that if the encoding of SNP1 and SNP2 does not sufficiently represent the marginal effects of SNP1 and SNP2 in the full model, the full regression model will force some of the trait variance to be explained by the interaction term (even if in truth there is no interaction effect at all) [32]. Any overfitting that could be at play by employing EDGE parameter $\alpha$ is accounted for in the main effects of the models. Our expectation of low inflation was validated in the conserved false positive rates demonstrated by EDGE in the simulation studies further described below. Thus, the EDGE encoding reduces the false positive interaction terms because it maximizes the information from the data to code marginal effects in the optimal manner.

To ensure that EDGE assigns the expected heterozygous genotype value across different types of underlying genetic models, we simulated main effect SNPs with the following genetic models using the Biallelic Model Simulator (BAMS) (for a description of BAMS, see Methods and S1 Text; further information and download available at https://www.hall-lab.org): recessive (REC), sub-additive (SUB), additive (ADD), super-additive (SUP), and dominant (DOM) (1,000 simulated datasets for each model type) across varying minor allele frequencies (MAFs) (S1 Fig). Here, heterozygous genotypes are simulated to have 0% (REC), 25% (SUB), 50% (ADD), 75% (SUP), and 100% (DOM) the risk of homozygous alternate genotypes (see Table 1). Fig 1 displays the distribution of the alpha values for each SNP model type at simulated 30% MAF. For every model, the density peaks correspond to the simulated genetic model (REC$\approx$0, SUB$\approx$0.25, ADD$\approx$0.5, SUP$\approx$0.75, and DOM$\approx$1), demonstrating that EDGE effectively assigns alpha values reflecting the simulated genetic model. Of note, a large portion (approximately half) of simulated recessive SNPs were assigned an alpha value between -0.5 and -0.001 and similarly, for simulated dominant SNPs, approximately half were assigned an alpha between 1.001 and 1.5. This is important to consider when natural SNPs are assigned alpha values below 0 or above than 1: while this could indicate a SNP's action is under-recessive or over-dominant, if the value is close to 0 or 1, the SNP's model may still be in the recessive or dominant range, respectively. When considering density peaks across MAFs, we observed variation in peak density across the allele frequencies (S1 Fig) with increased density at the expected alpha values for higher MAFs and greater variability for lower MAF (15% MAF or lower). One exception to this trend was for the simulated recessive acting SNPs, which showed the highest density for SNPs simulated with 5% MAF. Additionally, we observed differences in which simulated models showed the highest peak density at different MAFs. For example, as seen in Fig 1, at 30% MAF, the highest peak density was observed for simulated sub-additive SNPs. For MAF of 45%, EDGE assigned an alpha with the highest peak density for SNPs simulated with an additive model, and as described previously, recessive SNPs showed high peak density at 0 alpha with 5% MAF (S1 Fig). These results indicate that, while SNPs were assigned alpha values by EDGE that were consistent with the simulated model across all REC, SUB, ADD, SUP, and DOM models, EDGE performed most optimally at assigning an alpha value for SNPs whose simulated heterozygous risk relative to homozygous alternate is close in value to the MAF of the SNP.

## The additive encoding demonstrates inflation of type I error

To assess type I error, two categories of main effect SNP-SNP models were simulated: One-SNP Main Effect and Two-SNP Main Effect. For One-SNP Main Effect models, only one SNP exhibited a main effect while the other did not have an effect on the phenotype. The main effect SNPs were simulated with ADD, DOM, REC, SUB, and SUP genetic models. An additional genetic model was included, the heterozygous (HET) model, in which heterozygous

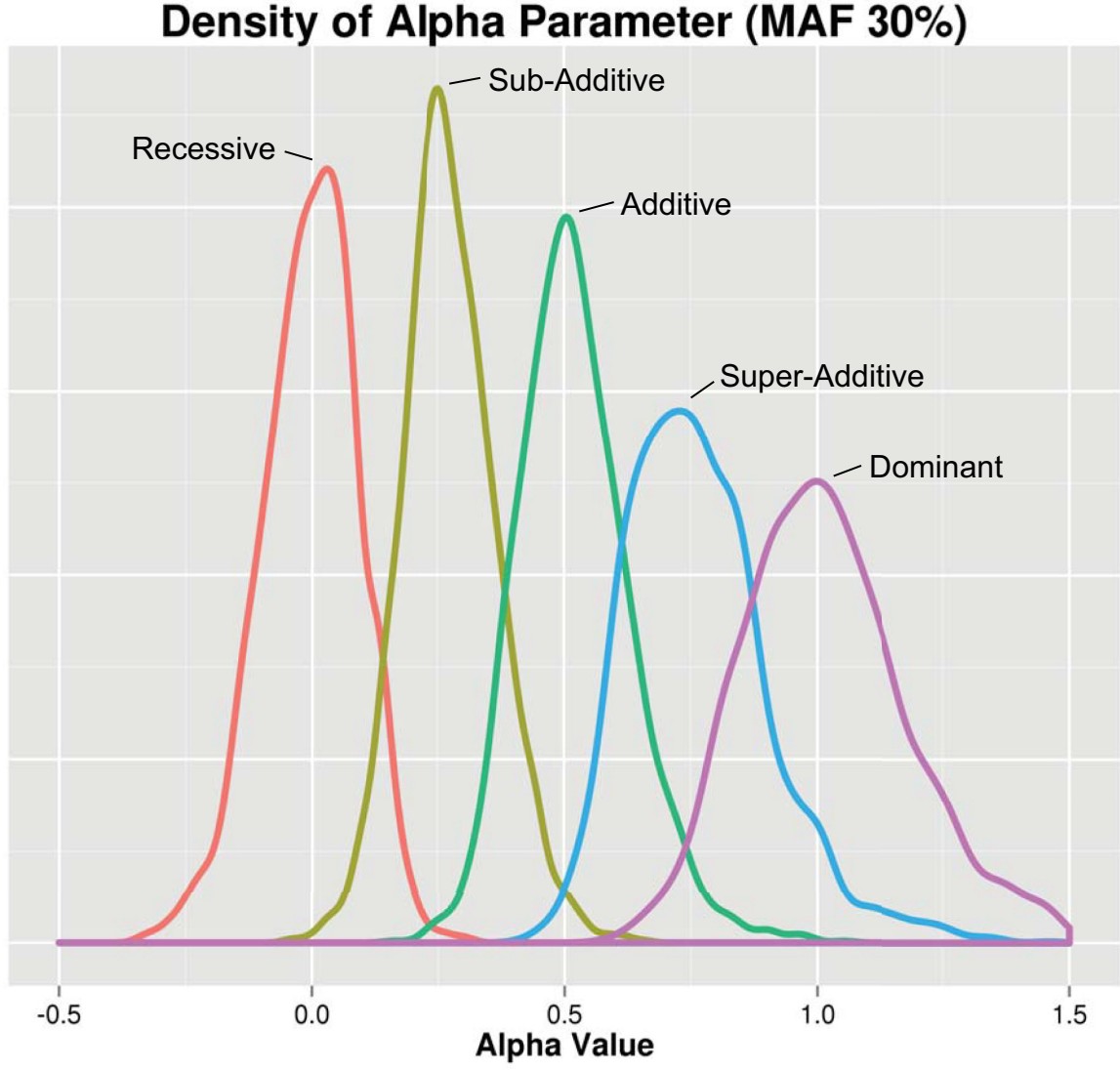

**Fig 1. Distribution of the estimated heterozygous action ($\alpha$).** Single SNP simulations were performed with five distinct underlying models: recessive (red), sub-additive (yellow), additive (green), super-additive (blue), and dominant (purple). All SNPs shown in this figure were simulated with a 30% MAF. 1,000 datasets for each SNP model were generated and the distribution of the alpha parameter was plotted. Along the x-axis is the EDGE assigned alpha value for simulated SNPs of each genetic model and along the y-axis is the density.

genotypes were simulated to have full risk relative to both homozygous genotypes, which had 0 risk. For Two-SNP Main Effect models, two SNPs with main effects were simulated with no interaction effect. We simulated all pairwise combinations between REC, SUB, ADD, SUP, DOM, and HET. Null samples with no main or interaction effects were also simulated. We simulated 1,000 datasets for each SNP-SNP model to calculate a false positive rate (FPR) as the percentage of the 1000 datasets with a likelihood ratio test (LRT) p-value < 0.05 per model using the additive, dominant, recessive, codominant, and EDGE encodings. FPR for the simulated main effect models represents the frequency at which the encoding identified a significant interaction term when no simulated interaction existed. As shown in Fig 2 with MAF of 30% as an example, all of the encodings demonstrated a FPR near 5% for the null data. For the main effect models, the majority of encodings showed conserved FPR. A deviation from this

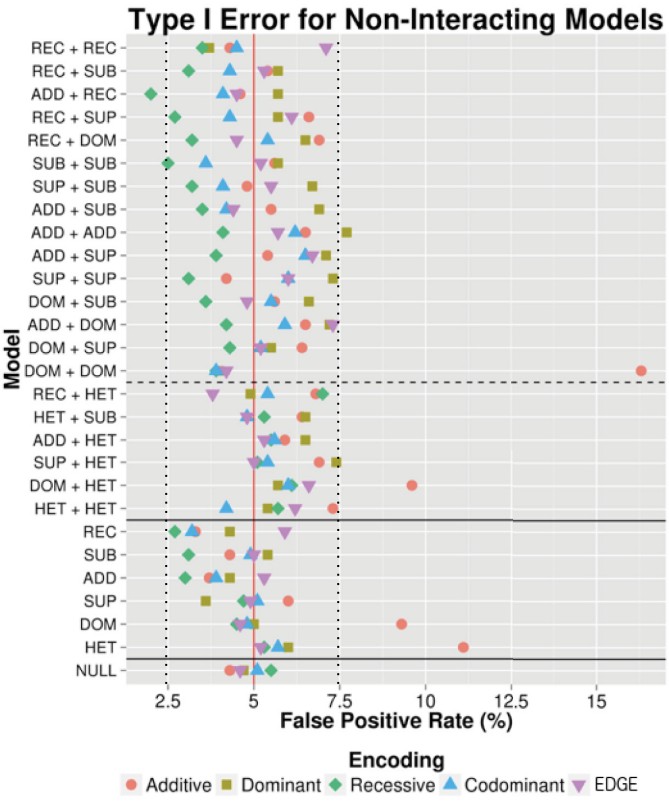

**Fig 2. Type I error in simulated main effect and null data.** Power to detect two-SNP (above the top solid horizontal line) and One-SNP (below top solid line) main effect simulations are displayed. Null model simulations are shown below the bottom solid horizontal line. The dashed horizontal line separates models involving HET SNPs (below dashed line) from non-HET SNPs (above the dashed lines) because the homozygous genotypes in HET SNPs were simulated to have 0 risk relative to the heterozygous genotype, while the homozygous alternate genotypes for all other SNPs were simulated to have full risk comparted to heterozygotes. Main Effect Only models and null data were simulated (1,000 datasets each) and LRT p-values were calculated using additive (red circle), dominant (yellow square), recessive (green diamond), codominant (blue triangle), and EDGE (purple inverted triangle) encodings to identify the percentage of time each encoding identified a significant interaction when there was none (false positive rate). The red vertical line marks a 5% false positive rate and the dashed black lines mark where the estimated type I error should fall within according to Bradley's liberal criteria [32]. All SNPs in this figure were simulated with a 30% minor allele frequency.

was seen for the additive encoding, where inflation was observed, including the scenarios under which: 1) both SNPs exhibit a main effect with a simulated dominant genetic model (FPR≅16%); 2) one SNP exhibits a main effect with a simulated heterozygous model (FPR≅11%), and 3) SNP1 exhibits a main effect with a simulated dominant model and SNP2 exhibits a main effect with a simulated heterozygous model (FPR≅9%). According to Bradley's liberal criteria [32], the estimated type I error should fall within the interval (0.025, 0.075). This is the case for the majority of encoding schemes except for additive encoding, which confirms the observation by Mahachie *et al* [32].

To compare the average false positive rate for the encodings among type I error datasets including the main effect and null datasets across 10%, 30%, and 50% MAF, we standardized signal to noise ratios. Fig 3 depicts the type I error at standardized, increasing signal to noise ratios using the different encodings. Each of the models from Fig 2 across the varying MAFs were averaged (solid lines in Fig 3). We observed low average type I error for the codominant

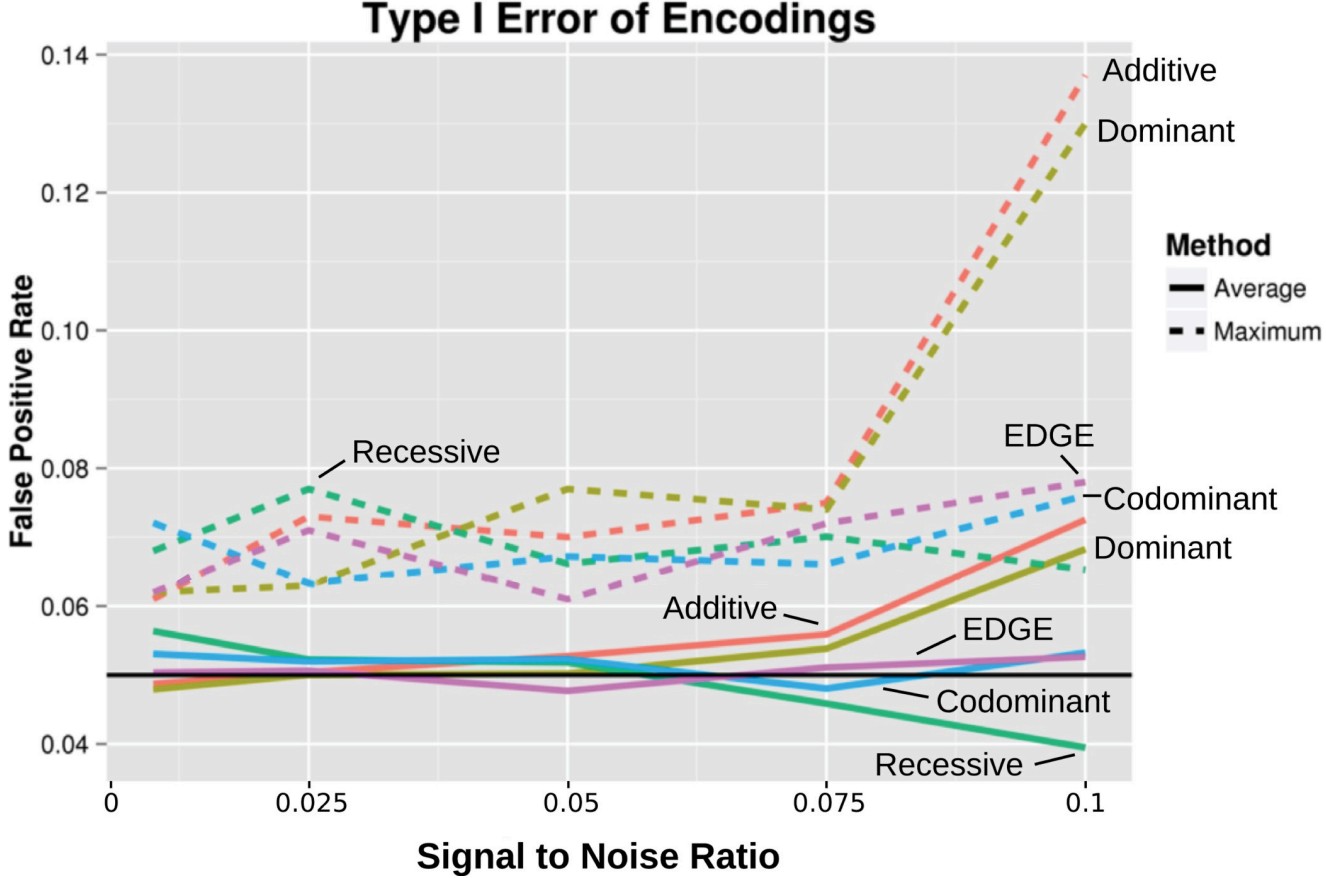

**Fig 3. Average and maximum type I error across all main effect and null simulated datasets.** Average false positive rates (solid lines) and the false positive rate for the maximum inflated model (dashed lines) were calculated for standardized signal to noise ratios. Values are plotted at increasing signal to noise ratios for each encoding: additive (red), dominant (yellow), recessive (green), codominant (blue), and EDGE (purple). The black line denotes a 5% false positive rate.

encoding and EDGE (following the 5% FPR level) as well as the recessive encoding (below the 5% level at high signal to noise ratios). However, at a higher signal to noise ratios, we again observed inflation of type I error for the additive and dominant encodings. Further inspection of each model's maximum type I error (dashed lines in Fig 3) at high signal to noise ratios revealed exceedingly high type I error for additive and dominant encodings.

## Power to detect interaction varies widely across genetic encoding and SNP model types

Once EDGE demonstrated accuracy at encoding individual SNPs based on the simulated genetic model and low type I error, we evaluated EDGE's power to detect SNP-SNP interaction models compared to the traditional encoding methods. We simulated 29 SNP-SNP interactions with no main effect. The first type of simulated SNP-SNP interaction models involved SNPs with comprehensive pairwise combinations of underlying genetic models: ADD, DOM, REC, SUB, SUP, and HET. Additionally, we evaluated simulated interactions using genotype-based interaction models that include penetrance functions (e.g., XOR, Hyperbolic) and scenarios in which only one of the 9 interaction penetrance cells confers risk while the other 8 demonstrate no risk (e.g., Homozygous Referent-Homozygous Referent–HR-HR). Power calculations for all 29 SNP-SNP interaction models using each of the encodings as the percentage of simulated datasets (out of

1,000) with an LRT p-value < 0.05. As an example, Fig 4 shows each encoding's power to detect the 29 interacting models we simulated, where MAF for both SNPs was 30%. Results demonstrated that additive, dominant, and recessive encodings outperformed the other encodings for the interaction models for which they are designed: ADDxADD, DOMxDOM, and RECxREC, respectively. EDGE showed over 80% power for every model in which at least one traditional encoding had over 80% power. The power to detect different models varied widely. For models in which one or two SNPs had an underlying REC model, no encoding identified signal with 80% or more power. Conversely, models including SNP(s) with DOM or SUP action demonstrated consistently high power across encodings. The recessive encoding demonstrated consistently low power and did not exceed 80% power for any model.

## EDGE demonstrates robust power compared to additive, dominant, recessive and codominant encodings

The power for an encoding to detect a given interaction model varies depending on the type of underlying genetic model (Fig 4). To assess average power for each encoding across the interacting models with consistency, signal to noise ratio was standardized across the different models. We compared the average power of each encoding across all traditional interacting models from Fig 4 and at increasing signal to noise ratios (10%, 30%, and 50% MAF). At 30% MAF (Fig 5), EDGE demonstrated the highest average power across traditional models with increasing signal to noise ratio; the codominant encoding showed comparable power to EDGE until the signal to noise reached a threshold at which increasing numbers of models diverged, reducing power; additive and dominant encodings demonstrated diminished power compared to EDGE; and recessive encoding showed low power to detect models even at high signal to noise ratio levels. Increasing the MAF to 50% resulted in greater average power for EDGE, codominant, and recessive, and reduced average power for dominant and additive when compared to 30% MAF. At 10% MAF, high average power for EDGE, additive, and dominant encodings was observed, while codominant and recessive showed exceedingly low power due to model non-convergence. Similar patterns were observed for the genotype-based models (Fig 6). However, the codominant encoding only demonstrated reduced average power for the datasets simulated with 10% MAF, and additive, dominant, and EDGE showed diminished power at 50% MAF as compared to Fig 5.

## Allele frequency and sample size influence power to detect models across all encodings

To test the impact of allele frequency, sample size, baseline risk, and case-control ratio on power, we performed a parameter test whereby all 29 interacting models were simulated with combinations of variation in each of the four parameters. We used ANOVA to test for significant effects of each parameter on the encodings' power to detect these simulated models. As shown in Fig 7, minor allele frequency and sample size demonstrated a significant effect on power for a large number of models for each encoding, while case-control ratio and baseline risk affected very few models. The differences observed across varying MAFs and sample sizes revealed similar trends: with increasing MAF and sample size, the power for each encoding to detect the interaction signal increased.

## EDGE assigns heterozygous values that are consistent with traditional encoding methods across five phenotypes

We applied EDGE to biomedical data from the Electronic Medical Records and Genomics (eMERGE) Network using age-related macular degeneration (AMD), age-related cataract, glaucoma, type 2 diabetes (T2D), and resistant hypertension.

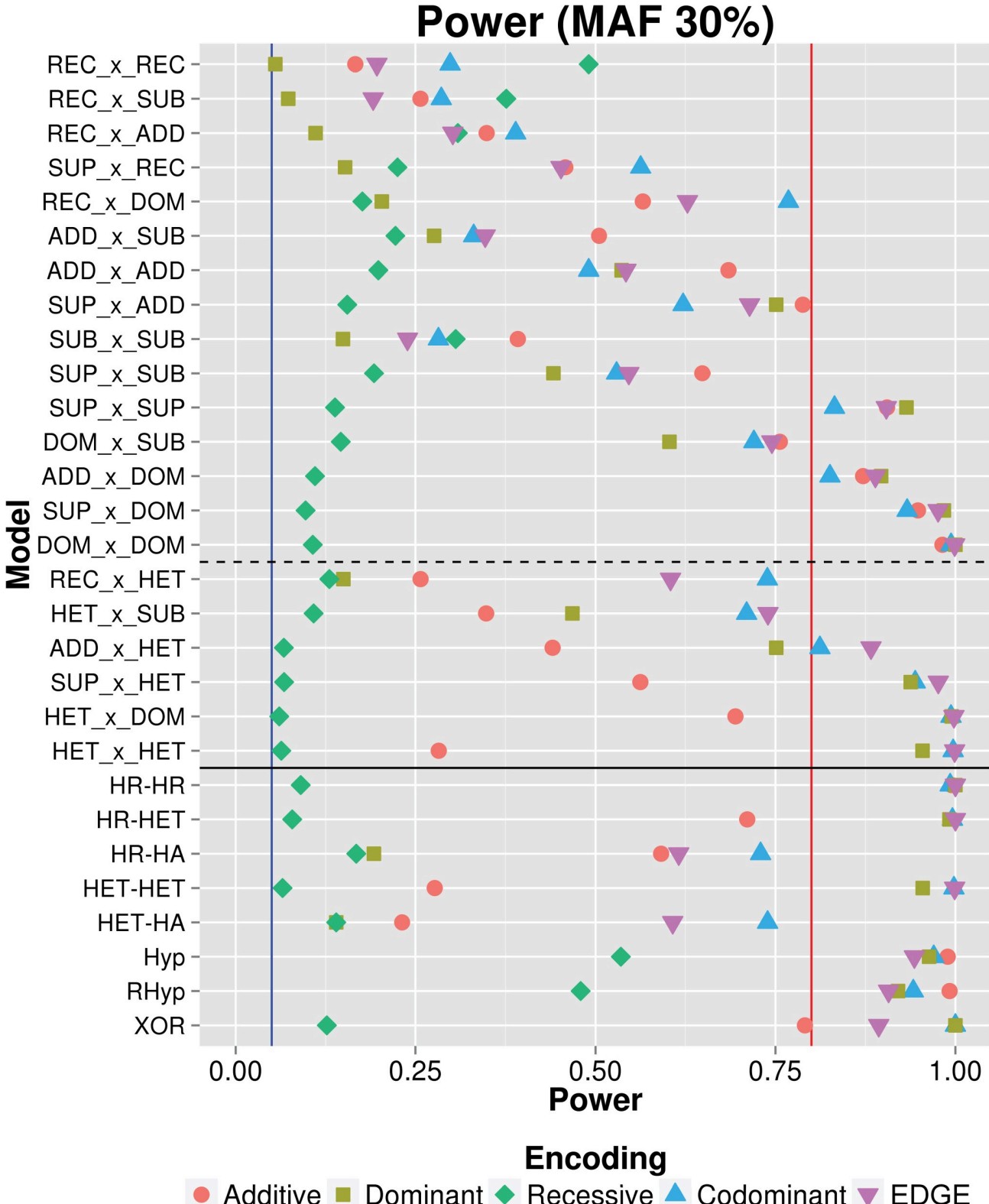

**Fig 4. Results of power analysis across 29 simulated SNP-SNP interaction models.** Power was calculated as the percentage of simulated models (out of 1,000 datasets for each model) for which the encoding detected an interaction signal at LRT p-value < 0.05: additive (red circle), dominant (yellow square), recessive (green diamond), codominant (blue triangle), and EDGE (purple inverted triangle). Twenty-nine SNP-SNP interaction models were

simulated. Above the solid horizontal line are interaction models with comprehensive two-SNP combinations between SNPs with REC, SUB, ADD, SUP, and DOM action (between the solid and dotted line are models including SNPs with HET action). Below the solid horizontal line are genotype-based interaction models. The blue and red vertical lines marks 5% and 80% power, respectively.

To see how EDGE-derived alpha values corresponded to traditional encoding main effect results, four GWAS were performed for each of the phenotypes (a multi-encoding GWAS), one for each traditional encoding: additive, dominant, recessive, and codominant. While genome-wide significant results were only found for T2D and AMD, all five phenotypes demonstrated similar patterns when the results of their five GWAS were plotted along the x-axis by each SNP's EDGE-derived heterozygous alpha value (Figs 8 and S2): For SNPs assigned an alpha value near 0, which indicates an underlying recessive genetic model, the recessive encoding tended to demonstrate the lowest p-values. Additive encoding predominated for SNPs with an alpha near 0.5 (indicating SNPs with an underlying additive model). Dominant encoding produced the lowest p-values for SNPs with an alpha near 1 (indicating SNPs with a dominant genetic model).

For T2D, 10 SNPs were identified at the genome-wide significance level ($5 \times 10^{-8}$) for at least one traditional encoding (Fig 8A and Sheets A-D in S1 Table). Six SNPs were significant for three out of the four traditional encodings, including the top three, and no SNP was genome-wide significant for all four encodings. rs4132670 (alpha: 0.68; MAF: 0.40; intron, *transcription factor 7 like 2 (TCF7L2)*) was the T2D top result with the additive encoding

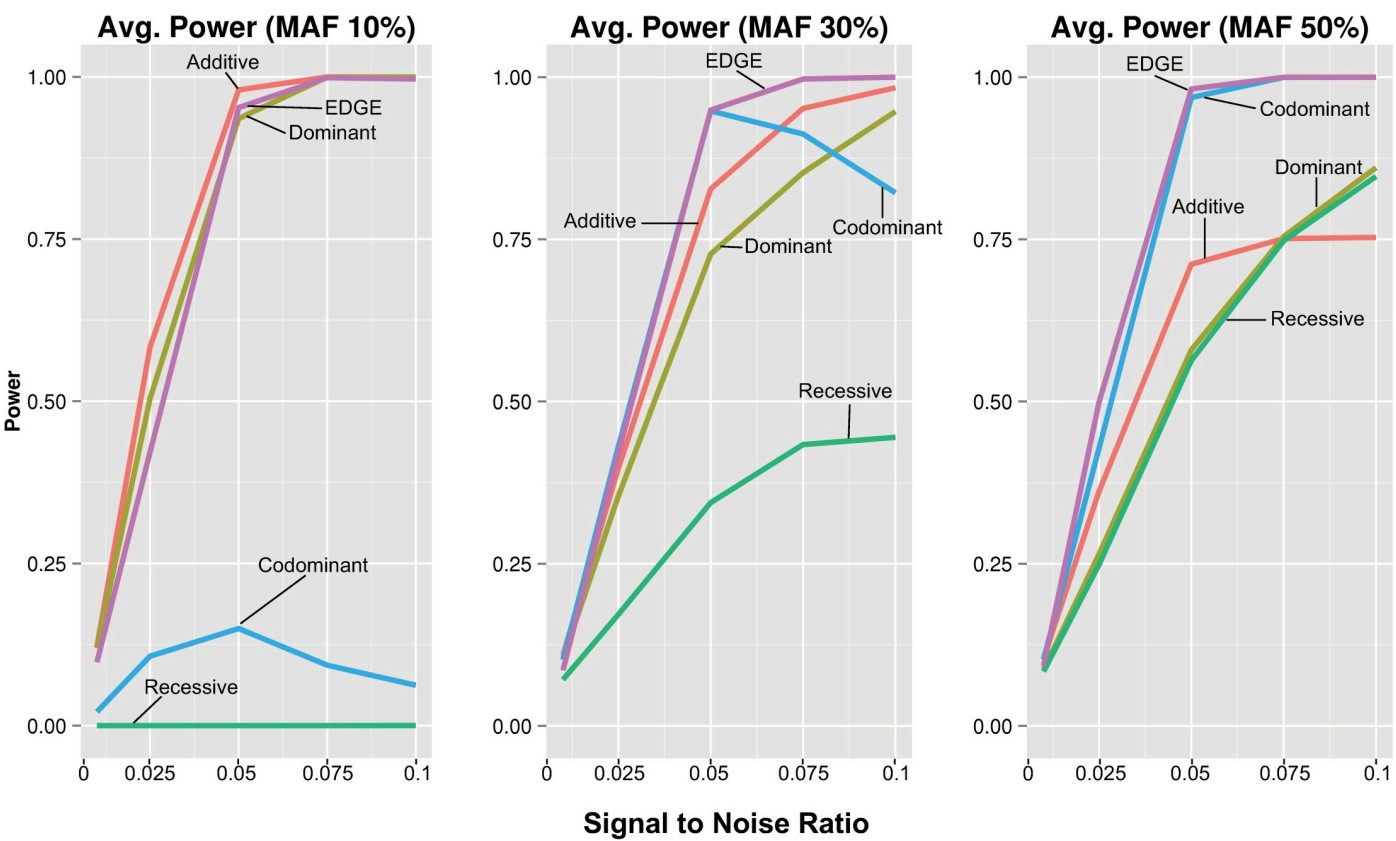

**Fig 5. Average power for standardized signal to noise ratio across all pairwise interaction models at 10%, 30% and 50% MAF.** Average power of each encoding was calculated for standardized signal to noise ratios and plotted by increasing signal to noise ratios for the pairwise model SNP-SNP interaction models using each encoding: additive (red), dominant (yellow), recessive (green), codominant (blue), and EDGE (purple).

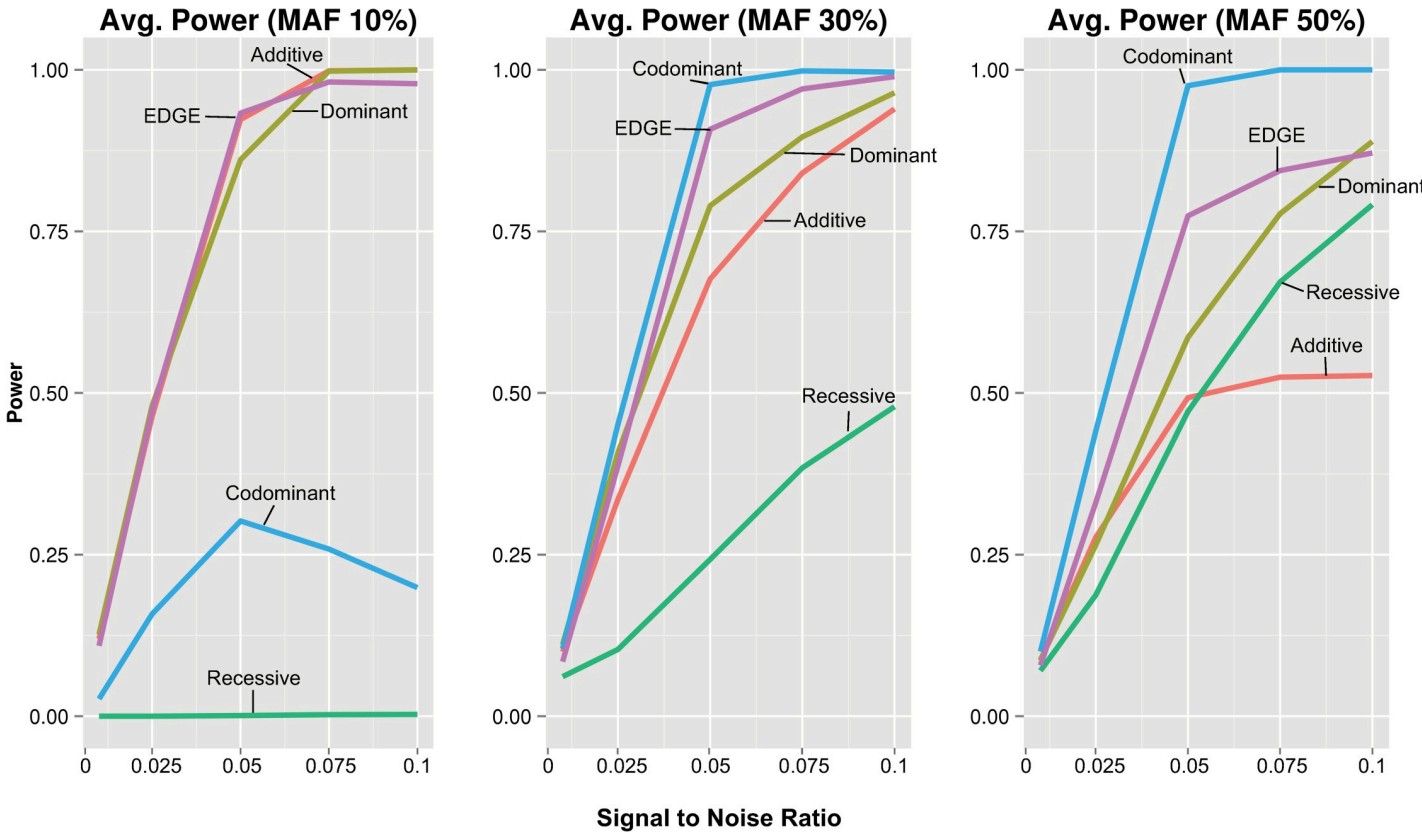

**Fig 6. Average power for standardized signal to noise ratio across all genotype-based interaction models at 10%, 30% and 50% MAF.** Average power was calculated for standardized increasing signal to noise ratios and plotted at increasing signal to noise ratios for the genotype-based SNP-SNP interaction models using each encoding: additive (red), dominant (yellow), recessive (green), codominant (blue), and EDGE (purple).

(Bonferroni p: $1.82 \times 10^{-13}$) followed by codominant encoding (Bonferroni p: $1.96 \times 10^{-13}$) and dominant encoding (Bonferroni p: $1.26 \times 10^{-13}$). It was not significant using the recessive encoding. rs12255372 (alpha: 0.53; MAF: 0.28; intron, *transcription factor 7 like 2 (TCF7L2)*) demonstrated the same order of significance for the encodings as rs4132670 ($1.45 \times 10^{-10}$, $1.50 \times 10^{-10}$, and $2.29 \times 10^{-8}$ for additive, codominant, and dominant Bonferroni adjusted p-values, respectively), and was also not genome-wide significant using the recessive encoding. The third top SNP result, rs2308953 (alpha: 0.13; MAF: 0.084; intron, *RAD1 checkpoint DNA exonuclease (RAD1)*), demonstrated the lowest p-value with the codominant encoding (Bonferroni p: $1.06 \times 10^{-5}$) followed by the recessive encoding (Bonferroni p: $3.39 \times 10^{-4}$) and additive (Bonferroni p: $9.93 \times 10^{-4}$) and was not genome-wide significant using the dominant encoding.

Eighteen SNPs were genome-wide significant for association with AMD using at least one of the traditional encoding methods (Fig 9 and Sheets A-D in S2 Table). Six SNPs were identified as genome-wide significant for all four traditional methods. The top result was for rs10801558 (alpha: 0.54; MAF: 0.41; intron, *complement factor H (CFH)*) using the codominant encoding (Bonferroni p: $8.21 \times 10^{-32}$), followed by additive encoding (Bonferroni p: $2.07 \times 10^{-31}$), dominant encoding (Bonferroni p: $5.41 \times 10^{-26}$), and recessive encoding (Bonferroni p: $2.73 \times 10^{-13}$). Three other SNPs in the *CFH* gene demonstrated genome-wide significance across all traditional encodings: rs399469 (alpha: 0.51; MAF: 0.46), rs10733086 (alpha: 0.50; MAF: 0.43;), and rs380390 (alpha: 0.50; MAF: 0.43;). Another top SNP was genome-wide significant for all four encodings, rs3750846 (alpha: 0.40; MAF: 0.23; *age-related maculopathy susceptibility 2*

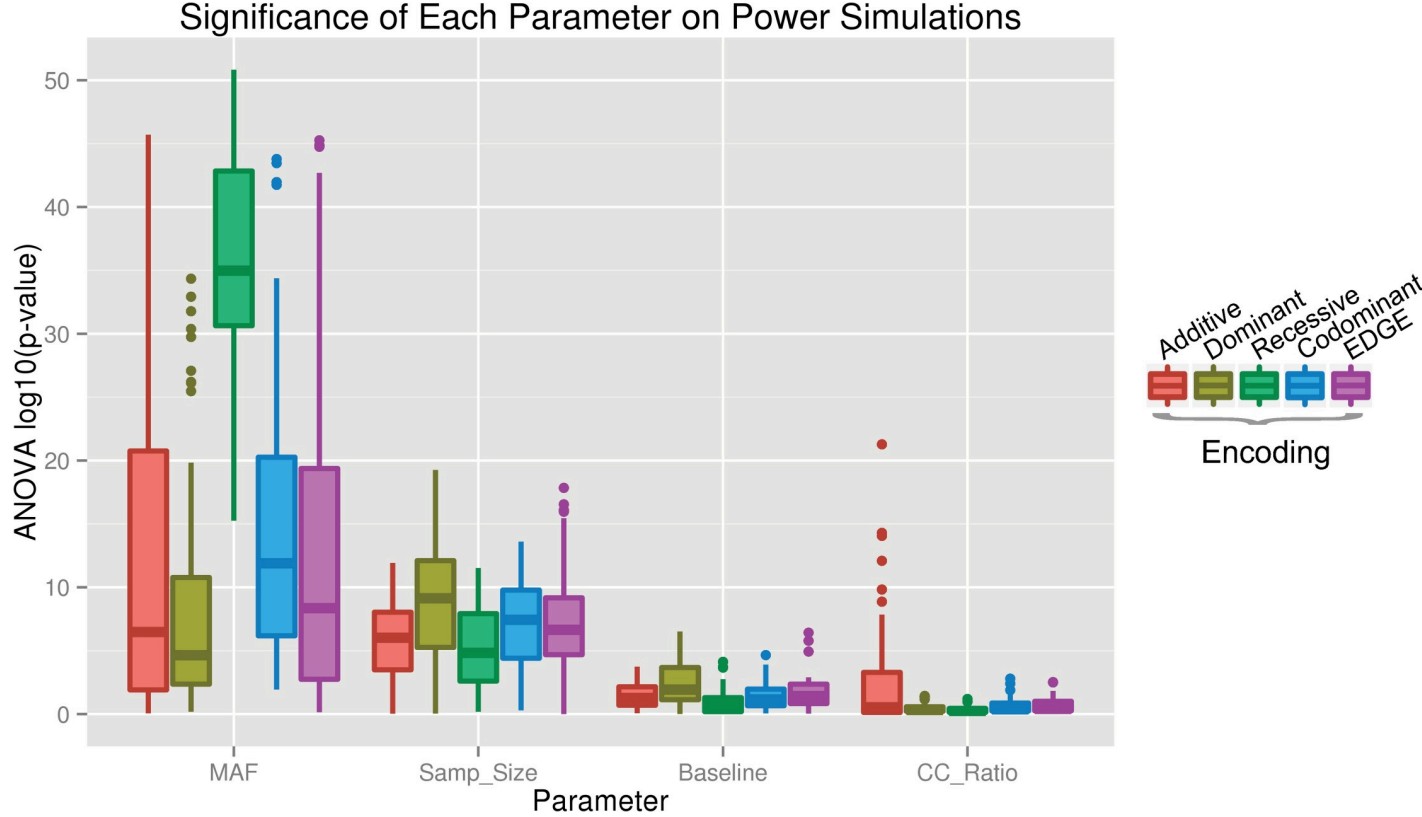

**Fig 7. Significance of each parameter on power.** ANOVA tests were performed to determine the impact of parameters MAF, sample size, baseline (penetrance), and case-control ratio on power for each interaction model for the additive (red), dominant (yellow), recessive (green), codominant (blue), and EDGE (purple) encodings.

(*ARMS2*)), with Bonferroni p: $5.76\times10^{-20}$ for additive, Bonferroni p: $2.47\times10^{-19}$ for codominant, Bonferroni p: $3.61\times10^{-14}$ for dominant, and Bonferroni p: $1.44\times10^{-9}$ for recessive. rs12042442 (alpha: 0.020, MAF: 0.15; intron of *assembly factor for spindle microtubules (ASPM)*) was genome-wide significant using the codominant encoding (Bonferroni p: 0.0035). This SNP was not genome-wide significant using the additive, recessive, and dominant encodings. No results from the multi-encoding GWAS for age related cataract (Sheets A-D in S3 Table), resistant hypertension (Sheets A-D in S4 Table), and glaucoma (Sheets A-D in S5 Table) met a genome-wide significance threshold.

To visualize the concordance of results between each pair of encodings for the multi-encoding GWAS for T2D and AMD, we generated pairwise scatterplots (Fig 9). Similar trends were seen for both phenotypes: we observed high concordance for -log10 p-values between codominant and additive, codominant and dominant, and additive and dominant (though for AMD additive and codominant encodings had higher -log10 p-values for some of the top SNPs compared to dominant). Recessive consistently showed lowered concordance with results from the other three encodings.

### Genetic interaction analysis of eMERGE results vary by phenotype and genetic encoding

After performing the multi-encoding GWAS for the five phenotypes, the results were used as a main effect filter for SNP-SNP interactions analysis. SNP-SNP interactions were assessed

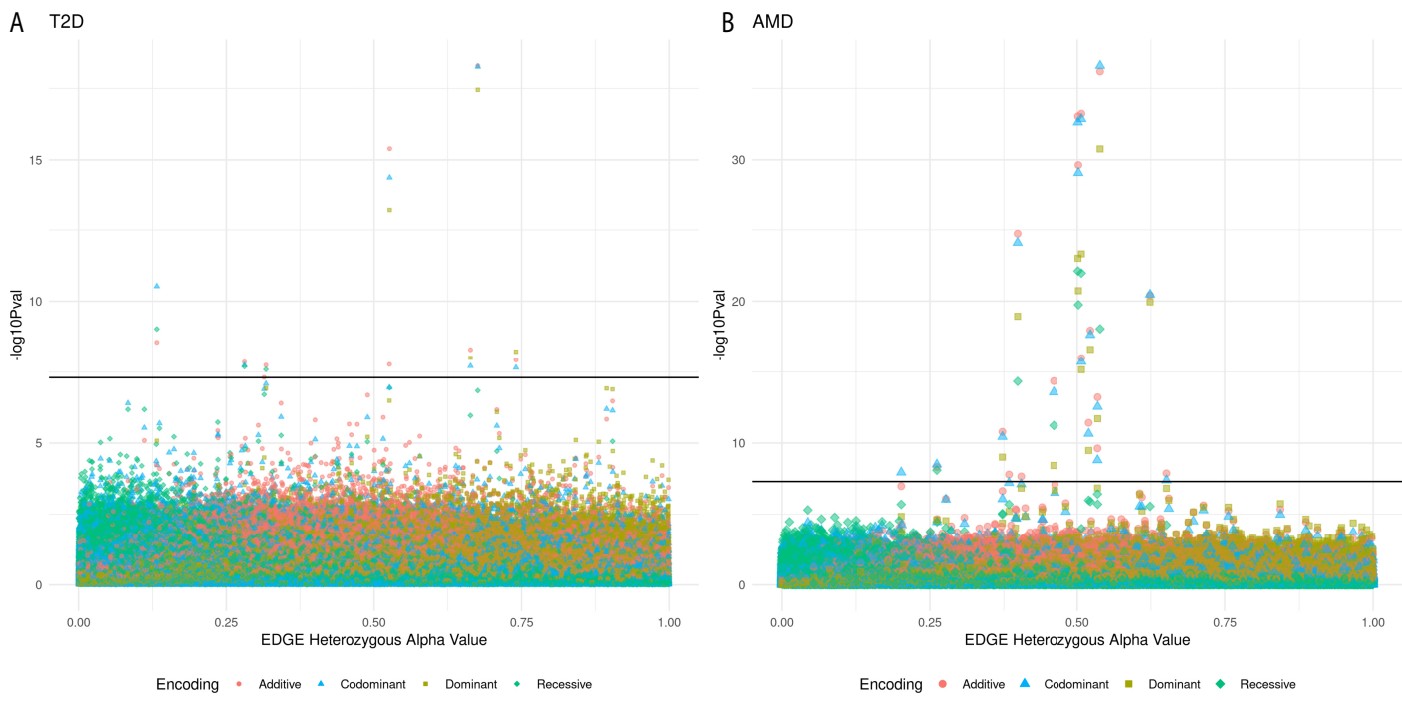

**Fig 8. Manhattan plot of results from a multi-encoding GWAS for T2D (A) and AMD (B).** GWAS were performed using the additive (red circle), dominant (yellow square), recessive (green diamond) and codominant (blue triangle) encodings for A) 358,569 SNPs and 20,341 samples (7,101 cases and 13,240 controls, 53% female) for type 2 diabetes and B) 311,161 SNPs and 13,153 samples (961 cases and 4,129, 56% female) for AMD. SNPs with EDGE-derived alpha values between 0 and 1 were sorted by the alpha value along the x-axis. Along the y-axis is the–log10 of the uncorrected p-value. The black line denotes the genome-wide significance threshold $(5 \times 10^{-8})$.

using the five encodings: additive, dominant, recessive, codominant, and EDGE by computing an LRT p-value (see Methods).

One SNP-SNP interaction model demonstrated a significant LRT p-value for T2D when adjusting for the number of tests: rs117537110 (alpha: 0.35; MAF: 0.45; intron, *protein phosphatase 1 regulatory subunit 18 (PPP1R18)*)–rs4149477 (alpha: -0.055; MAF: 0.48; intron, *tyrosylprotein sulfotransferase 2 (TPST2)*) using the recessive encoding (Bonferroni adjusted LRT p: 0.00051; 5,671 SNP-SNP models tested; $r^2$: 0.00015) (Fig 10 and Sheets A-E in S6 Table). Although it did not meet Bonferroni significance, EDGE identified this interaction with the next lowest p-value compared to the other methods (unadjusted LRT p: $8.91 \times 10^{-5}$). To determine if the SNP-SNP interaction models replicated in a separate dataset, we used data from the UK Biobank (UKB). Interaction replication analysis of these SNP pairs in UKB yielded no significant results for any encoding. We further evaluated potential replication of SNPs within a 50kb region of the original SNPs, and no significant epistasis models were identified for any encoding in this region-based replication analysis after multiple test correction.

For age-related cataract, one SNP-SNP interaction met a Bonferroni corrected significance threshold, rs7787286 (alpha: -0.037; MAF: 0.041; intergenic region of chromosome 7)–rs4695885 (alpha: 0.62; MAF: 0.34; intergenic region of chromosome 4), and this was found using EDGE (Bonferroni LRT p: 0.018; 9,591 SNP-SNP models tested; $r^2$: 0.00056) (Fig 11 and Sheets A-E in S7 Table). Interaction between these two SNPs was not replicated in UKB, where the MAFs were 0.030 and 0.34 for rs7787286 and rs4695885, respectively. The recessive encoding identified signal in a region-based interaction replication in UKB between SNPs

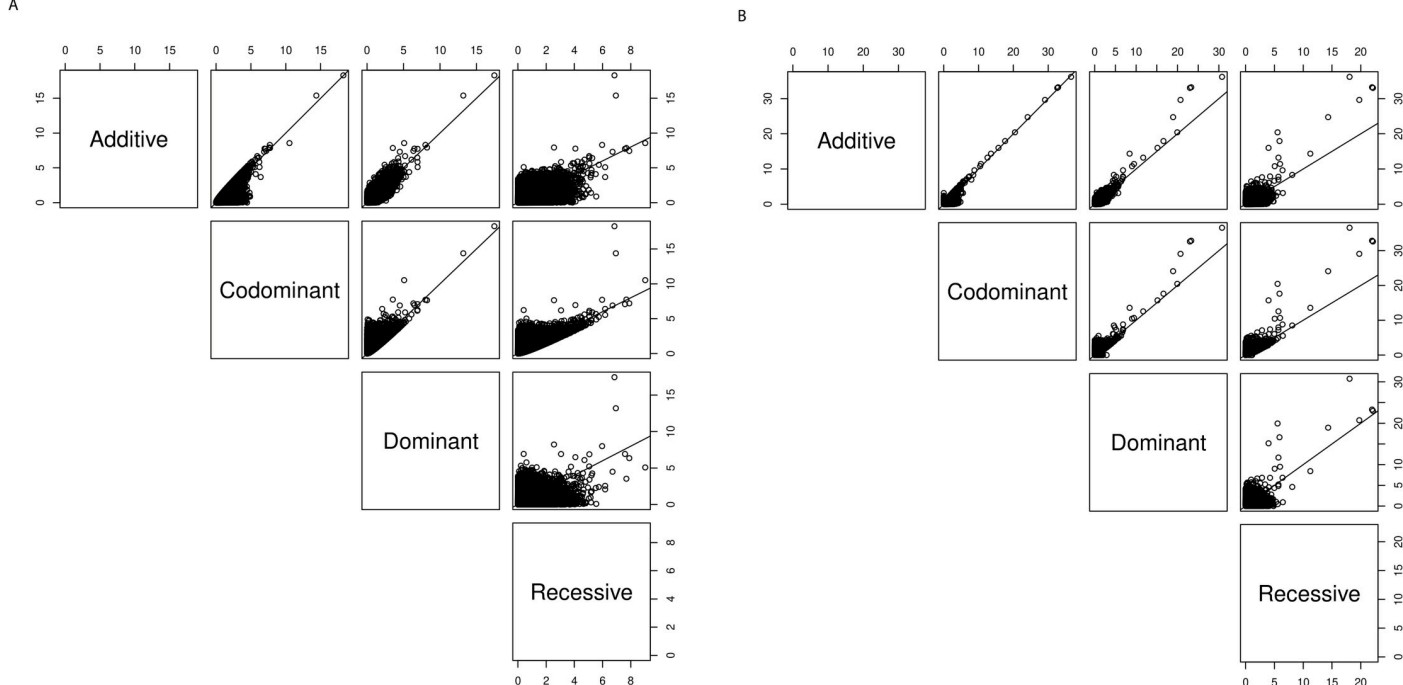

**Fig 9. Pairwise comparisons of the -log10 p-values for the multi-encoding GWAS of (A) T2D and (B) AMD.** For each SNP, we plotted the -log10 p-values for pairwise combination of encodings in six scatterplots for (A) T2D and (B) AMD compared to the identity line (1:1 line) such that pairs of encodings having p-values that fall closer to the line are more concordant.

rs60374751 (MAF: 0.030 and alpha: -0.65; 17,434 bp from rs7787286) and rs6843594 (MAF: 0.34 and alpha: -0.54; 24,084 bp from rs4695885) (Bonferroni LRT p: 0.026; $r^2$: 0.00000033) (Sheets A-E in S8 Table).

One SNP-SNP model met a corrected significance threshold for resistant hypertension using the dominant encoding: rs3801888 (alpha: 0.028; MAF: 0.28) in *sorting nexin 10 (SNX10)* and rs2858808 (alpha: -1.39; MAF: 0.28) (Bonferroni adjusted LRT p: 0.047; 10,296 SNP-SNP models tested; $r^2$: 0.00037) (Fig 12 and Sheets A-E in S9 Table). The resistant hypertension phenotype was not available in UKB for replication. We tested an interaction between these SNPs for hypertension in UKB due to its relation to the resistant hypertension phenotype and availability in the dataset. A significant interaction was found between rs3801888 (MAF: 0.27 and alpha: 1.097) and rs2858808 (MAF: 0.30 and alpha: 0.23) in UKB using the recessive encoding (LRT p: 0.025; $r^2$: 0.0000021) (Sheets A-E in S10 Table).

In the AMD interaction analysis, two SNP-SNP interaction models were significant when adjusting for the number of tests run (4,278 SNP-SNP models tested) (Fig 13 and Sheets A-E in S11 Table). The top SNP-SNP model for this phenotype was rs2336502 (alpha: 0.39; MAF: 0.33; pseudogene LOC100996886)–rs6695321 (alpha: 0.52; MAF: 0.36; intron, *CFH*), which was identified by the additive encoding (Bonferroni LRT p: $4.3 \times 10^{-12}$), EDGE encoding (Bonferroni LRT p: $6.9 \times 10^{-12}$), codominant encoding (Bonferroni LRT p: $6.8 \times 10^{-10}$), dominant encoding (Bonferroni LRT p: $3.2 \times 10^{-8}$), and recessive encoding (Bonferroni LRT p: 0.026) ($r^2$: 0.0030). Another significant interaction model also included rs2336502, which was identified as interacting with rs5993 (alpha: 0.28; MAF: 0.26; intron, *coagulation factor XIII B chain (F13B)*) using the additive encoding (Bonferroni LRT p: 0.021) and EDGE (Bonferroni LRT p: 0.047) ($r^2$: 0.00010). In the UKB replication, the interaction between rs2336502 (alpha: 0.55; MAF: 0.31) and rs6695321 (alpha: 0.52; MAF: 0.38) was significant (Sheets A-E in S12 Table)

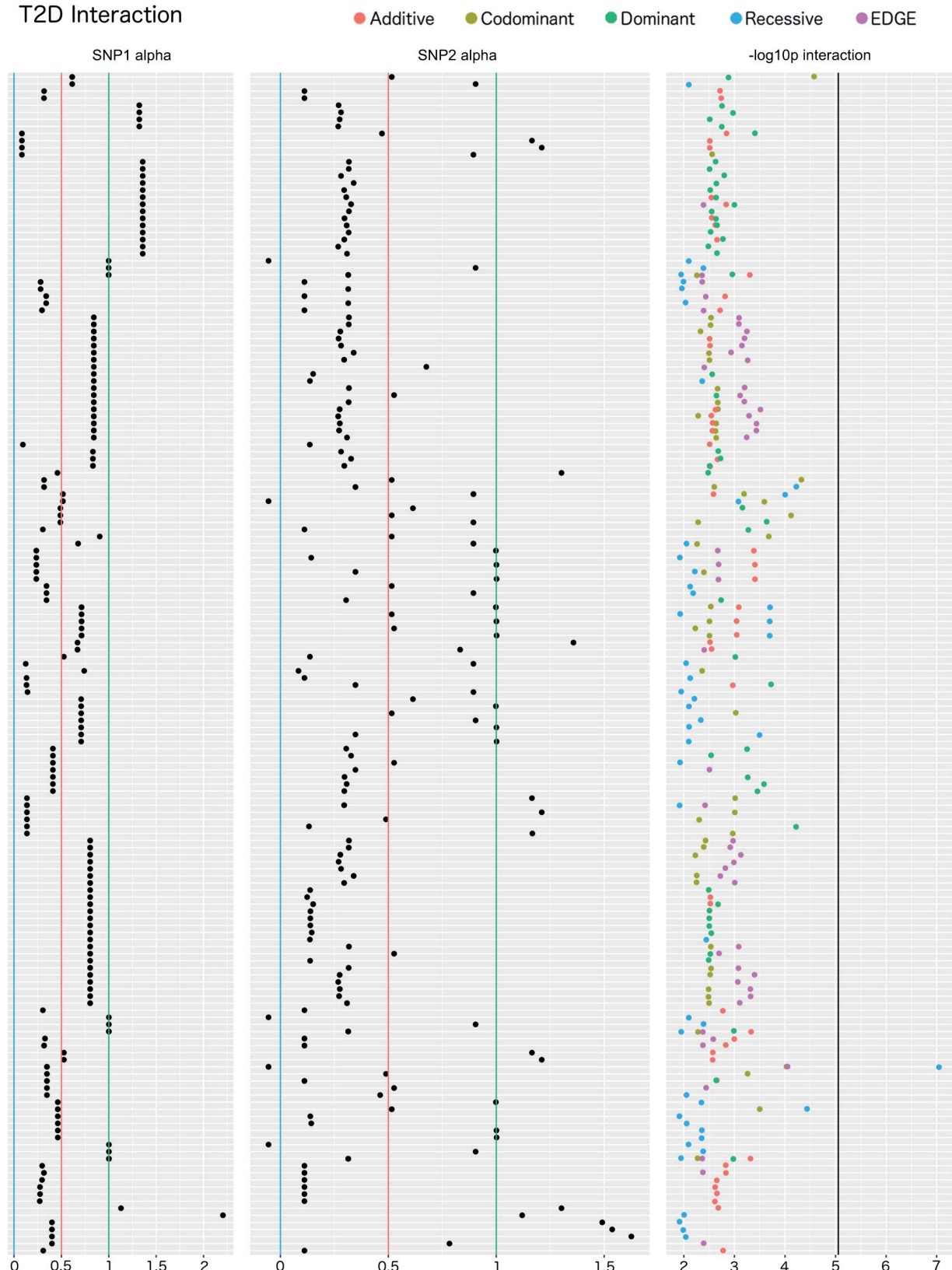

**Fig 10. Interaction plot of the top results from the main effect filtered SNP-SNP interaction analysis for T2D.** The top 50 SNPs from the multi-encoding T2D GWAS for each traditional encoding were considered for a pairwise SNP-SNP interaction analysis using additive (red), dominant (green), recessive (blue), codominant (yellow) and EDGE (purple) encoding methods. Track one displays the EDGE-derived heterozygous alpha value of SNP 1, track 2 displays the alpha value for SNP 2, and track 3 displays the–log10 of the unadjusted LRT p-value of each SNP-SNP interaction model. Vertical blue lines denote a 0 alpha value (indicative of recessive genetic model), vertical red lines denote a 0.5 alpha value (indicative of additive genetic model), vertical green lines denote an alpha value of 1 (indicative of dominant genetic model). Note that some of the SNPs demonstrated alpha values outside the 0–1 range, indicative of under-recessive ($\alpha < 0$) and over-dominant ($\alpha > 1$) genetic models. The vertical black line denotes the Bonferroni significance threshold for this analysis (5,671 SNP-SNP models tested).

for the additive (LRT p: $1.7 \times 10^{-4}$), codominant (LRT p: 0.0021), dominant (LRT p: $2.4 \times 10^{-4}$), and EDGE (LRT p: $1.2 \times 10^{-4}$) encodings ($r^2$: 0.000047). Replication interaction for SNPs rs2336502 and rs5993 (alpha: 0.60; MAF: 0.16) was significant for the dominant encoding (LRT p: 0.016; $r^2$: 0.040). There were no interaction results that met a Bonferroni corrected LRT p-value threshold for the interaction analysis of glaucoma.

## Discussion

For over a decade, the additive model has been the most common method for encoding SNPs in regression-based epistasis. In this paper, we aimed to introduce a novel encoding that is flexible to detect SNPs with nonadditive allelic architecture, evaluate the different genetic encodings in the context of epistasis, find evidence that some SNPs may demonstrate a nonadditive model, and identify novel SNP-SNP interactions associated with complex disease.

The elastic data-driven genetic encoding (EDGE) is a novel, robust alternative to the traditional methods for encoding genotypes. EDGE assigns a heterozygous genotype with a unique value based on each SNP's heterozygous risk relative to its homozygous alternate risk. EDGE accurately assigned heterozygous values to simulated SNPs based on their unique underlying genetic models, and maintained low type I error. However, the additive and dominant encodings demonstrated inflation. Therefore, a portion of results published using the additive model for epistasis could be false positives. Specifically, our results indicate that the additive encoding can falsely identify SNPs that exhibit main effect dominant action as having an interaction that predicts the outcome above and beyond the main effects when there is no interaction effect.

We applied EDGE to simulated epistasis models and compared its performance to the traditional methods. As expected, additive, dominant, and recessive encodings demonstrated power to detect the types of models for which they were designed. Despite their success in these scenarios, the methods varied widely in their power to detect other model types. Conversely, EDGE demonstrated robust power across numerous types of models and was among the highest for average power with every minor allele frequency tested. Additionally, when compared to the codominant encoding, EDGE retained sufficient power for situations in which the codominant encoding lost power (30% and 10% MAF and high signal to noise ratios). These findings suggest that EDGE is a flexible method to detect epistasis signal for SNPs when the genetic model is unknown.

The multi-encoding GWAS results of data from the eMERGE Network for T2D, AMD, age-related cataract, resistant hypertension, and glaucoma revealed similar patterns across the phenotypes when sorted by the EDGE-derived heterozygous alpha value: the encodings demonstrating the top p-values for SNPs tended to corroborate the alpha value that EDGE assigned. Top results changed depending on the encoding employed for this main effect analysis. The top three SNPs in the results of the multi-encoding T2D GWAS were not identified by all traditional encoding methods. Two of these SNPs (rs4132670 and rs12255372) are intronic variants in *TCF7L2* and have both been previously identified for association with type 2 diabetes [33] as well as body mass index [34] and retinopathy [33]. Both of these SNPs had alpha

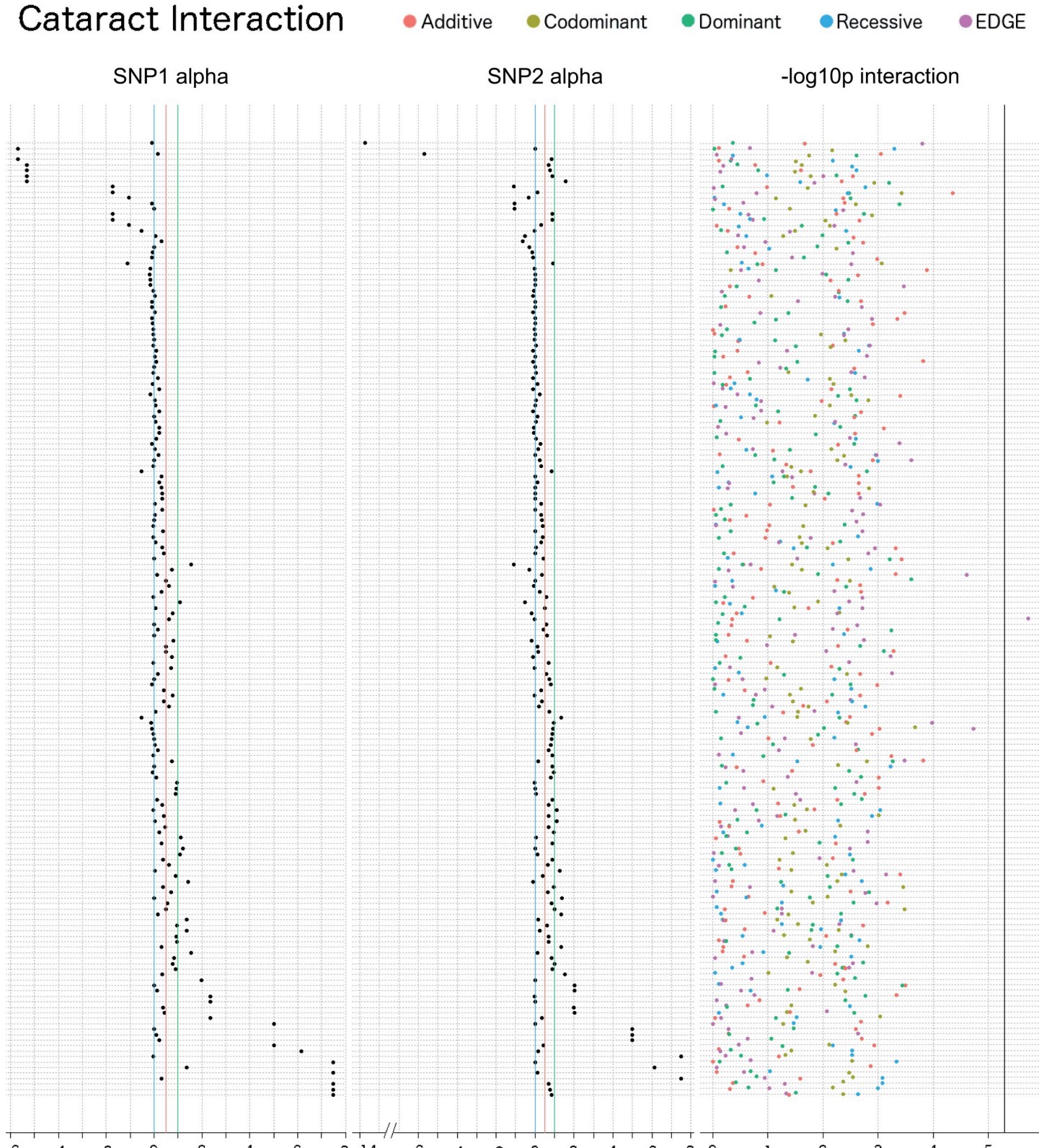

**Fig 11. Interaction plot of the top results from the main effect filtered SNP-SNP interaction analysis for age-related cataract in the eMERGE Network.** The top 50 SNPs from the multi-encoding cataract GWAS in the eMERGE Network were considered for a pairwise SNP-SNP interaction analysis using additive (red), dominant (green), recessive (blue), codominant (yellow) and EDGE (purple) encoding methods. Track one displays the EDGE-derived heterozygous alpha value of SNP 1, track 2

displays the alpha value for SNP 2, and track 3 displays the–log10 of the unadjusted LRT p-value of each SNP-SNP interaction model. Vertical blue lines denote a 0 alpha value (indicative of recessive genetic model), vertical red lines denote a 0.5 alpha value (indicative of additive genetic model), vertical green lines denote an alpha value of 1 (indicative of dominant genetic model). Note that some of the SNPs demonstrated alpha values outside the 0–1 range, indicative of under-recessive ($\alpha < 0$) and over-dominant ($\alpha > 1$) genetic models. The vertical black line denotes the Bonferroni significance threshold for this analysis (9,591 SNP-SNP models tested).

values between 0.5 and 0.75 (0.68 for rs4132670 and 0.53 for rs12255372), indicating an underlying model between additive and super-additive for both, and they were genome-wide significant using additive, codominant, and dominant encodings but not recessive. Another top SNP for T2D, rs2308953, in *RAD1*, was assigned an alpha of 0.13, indicating an underlying recessive model. This SNP demonstrated genome-wide significance for the codominant, recessive, and additive encodings, but was not significant using the dominant encoding. To the authors' knowledge, this is the first association between T2D and this SNP as well as this gene.

In contrast to the results from the T2D GWAS, the multi-encoding GWAS for AMD had more consistency in results across the encodings. Of the 18 SNPs with a genome-wide significant hit from at least one encoding, 6 were identified by all of the traditional methods. Four of these SNPs are in the *CFH* gene, a gene with a well-documented link to AMD [5,35–38] and each of these SNPs were assigned alpha values between 0.50 and 0.54, indicating an additive genetic model. One SNP, rs12042442, in the intron of *ASPM*, was assigned an alpha value of 0.020, reflective of a sub-additive genetic model, and was only significant using the codominant encoding. This SNP has no known previous associations, while other SNPs in *ASPM* have been reported to be associated with AMD [20] and end-stage coagulation [39].

Genetic interaction analyses in eMERGE yielded significant results for four of the five phenotypes. In T2D, one SNP-SNP interaction between rs117537110 (alpha: -0.055) and rs4149477 (alpha: 0.35) met the Bonferroni-corrected significance criterion using the recessive encoding, while no other encodings identified a statistically significant result. Considering the low power that the recessive encoding demonstrated in our simulation results, it is notable that recessive was the only encoding to identify a significant interaction for this phenotype. Yet, the EDGE-derived alpha with which each SNP was assigned indicate recessive and sub-additive underlying genetic models, which may explain why the recessive encoding was able to identify an interaction between these two SNPs, and our simulation power simulations demonstrated that each traditional encoding outperform other encodings when the model involved one or more SNPs they were designed to detect. These SNPs and the genes in which they reside, *TPST2* and *PPP1R18*, have no previously demonstrated associations with T2D, perhaps because the recessive encoding is not the typical method of choice for genetic associations and interaction analysis. These SNPs did not replicate in data from the UKB for any encoding in both the exact SNP-SNP interaction or a region-based interaction replication analyses.

One SNP-SNP interaction model, rs3801888 (alpha: 0.028; *SNX10*) and rs2858808 (alpha: -1.39; *STARD13*), was identified by the dominant encoding for resistant hypertension. The SNPs were assigned alpha values indicative of recessive and under-recessive genetic models. This result offers an opportunity for further investigation as this type of biological interaction model was not assessed in the simulation experiments. Future work will be needed to ensure that no inflation of interaction p-value is observed using the dominant encoding for two main effect SNPs with these underlying models. While the resistant hypertension phenotype was not available in UKB, hypertension was, and the recessive encoding identified an interaction between these SNPs associated with hypertension. In UKB, rs3801888 was assigned an alpha of 1.097, indicative of a dominant genetic model for its association with hypertension, and rs2858808 was assigned an alpha of 0.23, indicating a sub-additive encoding. Though we do not consider this to be a replication because these are distinct phenotypes, the results indicate

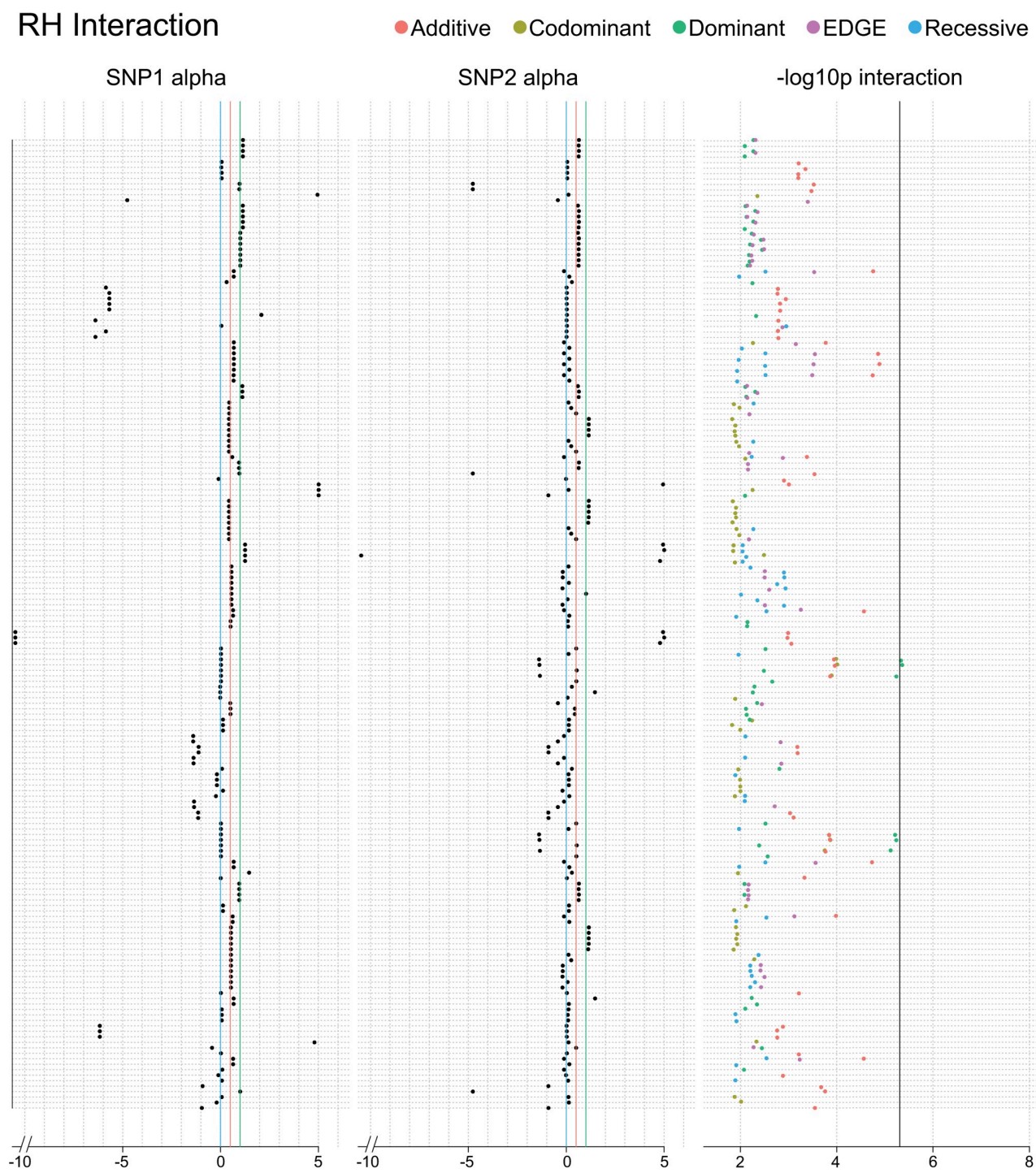

**Fig 12. Interaction plot of the top results from the main effect filtered SNP-SNP interaction analysis for resistant hypertension.** The top 50 SNPs from the multi-encoding resistant hypertension GWAS were considered for a pairwise SNP-SNP interaction analysis using additive (red), dominant (green), recessive (blue), codominant (yellow) and EDGE (purple) encoding methods. Track one displays the EDGE-derived heterozygous alpha value of SNP 1, track 2 displays the alpha value for SNP 2, and track 3 displays the–log10 of the unadjusted LRT p-value of each SNP-SNP interaction model. Vertical blue lines denote a 0 alpha value (indicative of recessive genetic model), vertical red lines denote a 0.5 alpha value (indicative of additive genetic model), vertical green lines denote an alpha value of 1 (indicative of dominant genetic model). Note that some of the SNPs demonstrated alpha values outside the 0–1 range, indicative of under-recessive ($\alpha < 0$) and over-dominant ($\alpha > 1$) genetic models. The vertical black line denotes the Bonferroni significance threshold for this analysis (10,296 SNP-SNP models tested).

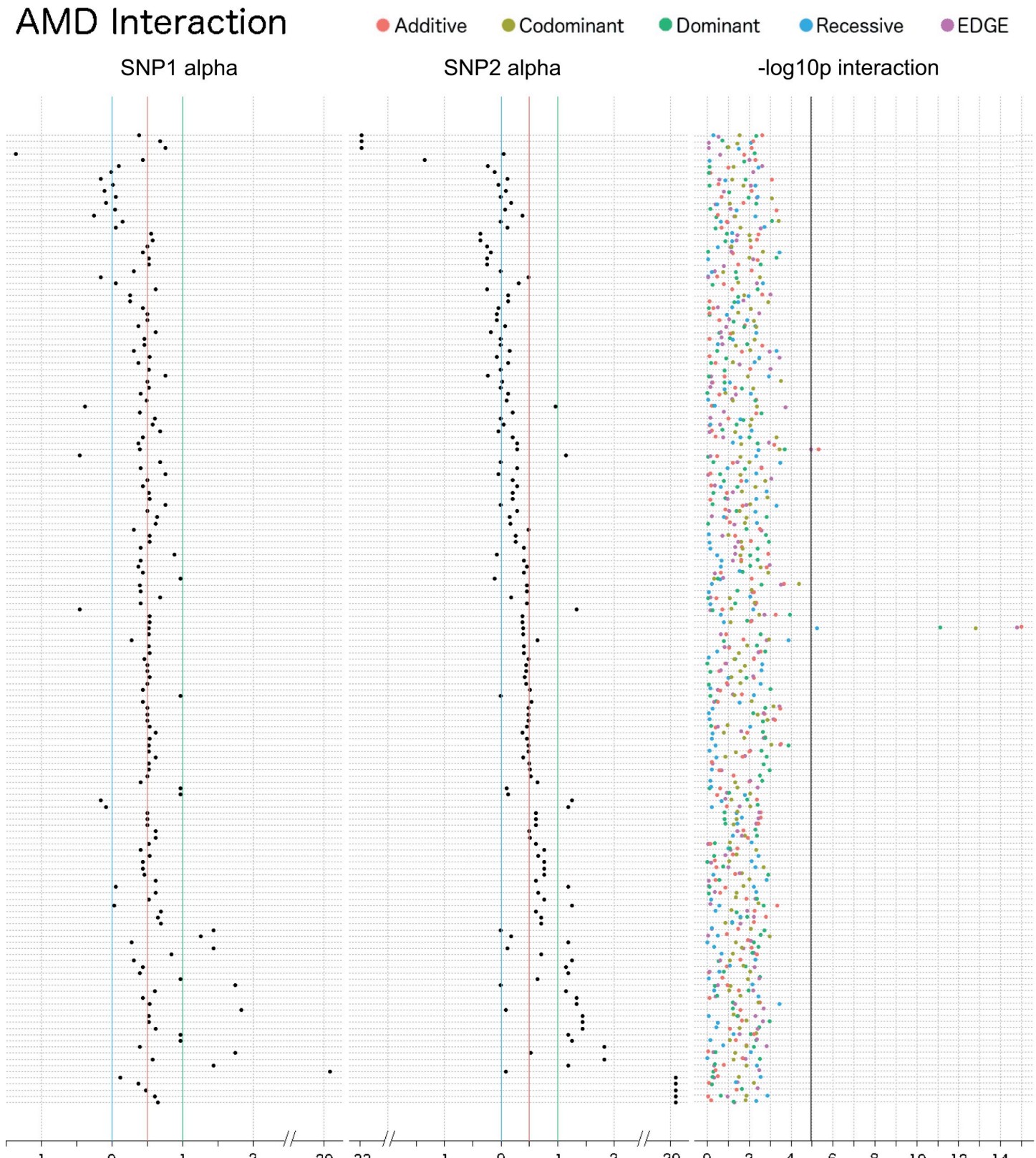

**Fig 13. Interaction plot of the top results from the main effect filtered SNP-SNP interaction analysis for AMD.** The top 50 SNPs from the multi-encoding AMD GWAS were considered for a pairwise SNP-SNP interaction analysis using additive (red), dominant (green), recessive (blue), codominant (yellow) and EDGE (purple)

encoding methods. Track one displays the EDGE-derived heterozygous alpha value of SNP 1, track 2 displays the alpha value for SNP 2, and track 3 displays the–log10 of the unadjusted LRT p-value of each SNP-SNP interaction model. Vertical blue lines denote a 0 alpha value (indicative of recessive genetic model), vertical red lines denote a 0.5 alpha value (indicative of additive genetic model), vertical green lines denote an alpha value of 1 (indicative of dominant genetic model). Note that some of the SNPs demonstrated alpha values outside the 0–1 range, indicative of under-recessive ($\alpha < 0$) and over-dominant ($\alpha > 1$) genetic models. The vertical black line denotes the Bonferroni significance threshold for this analysis (4,278 SNP-SNP models tested).

a potential similarity in the genetic mechanisms involved in these two related phenotypes, and the difference of alpha assignments for the SNPs may indicate a complex interrelationship between the two phenotypes that warrants further study.

Using EDGE, one SNP-SNP interaction model involving two intergenic SNPs, rs7787286 (alpha: -0.037) and rs4695885 (alpha: 0.62), was identified as significant for association with age-related cataract. Each SNP demonstrated unique alpha values: one indicative of a recessive/under-recessive genetic model, while the other SNP's indicated an additive genetic model. The ability of EDGE to identify these interacting SNPs demonstrates the flexibility of the method to identify interactions between SNPs with different underlying genetic models. rs4695885 has demonstrated an association with age-related cataract from a previous GWAS [40], but the result did not reach genome-wide significance in the previous study. rs7787286 has no known previous associations. Though an interaction between these exact SNPs was not found for any encoding in UKB, the recessive encoding identified signal in the region-based interaction in UKB between SNPs rs60374751 (alpha: -0.65; 17,434 bp from rs7787286) and rs6843594 (alpha: -0.54; 24,084 bp from rs4695885). Notably, these SNPs' alpha values are different from the original SNPs. As previously described, our simulation results show that the encoding that has the best power is the one designed for the genetic model demonstrated by the SNPs in the interaction model, and both of these SNPs appear to have a recessive/under-recessive genetic model, according to EDGE. While EDGE first identified this interaction in eMERGE, the recessive encoding likely had higher power to detect an interaction between these SNPs in UKB. This is the first known association for both SNPs identified in UKB.

Two SNP-SNP models were significant for AMD and both included rs2336502 in a pseudo-gene region *LOC100996886* with an alpha of 0.39 (indicating a genetic model between sub-additive and additive) with no known previous associations. This SNP was found to interact with rs6695321 in the intron of *CFH* with an alpha of 0.52 (indicating an additive genetic model) and was identified by all five encoding methods. As previously discussed, *CFH* has well-established associations with AMD, and rs6695321 has previous association with AMD as well [41]. The other SNP found to interact with rs2336502 (using EDGE and the additive encodings) was rs5993 (alpha: 0.28, indicating sub-additive genetic model; *F13B*), and while *F13B* has previous association with AMD [20] this SNP has no known previous associations. In the UKB replication, the interaction between rs2336502 (alpha: 0.55) and rs6695321 (alpha: 0.52) was significant for all encodings except recessive. Replication interaction for rs2336502 and rs5993 (alpha: 0.60) was significant for the dominant encoding. With the exception of rs5993, the alpha values were fairly consistent for the SNPs in the two datasets.

The results of the simulation study and epistasis analysis in data from the eMERGE Network and UK Biobank indicate that there are examples of SNPs with a nonadditive genetic model, which demonstrates the need for encoding methods beyond additive, including EDGE. Some limitations in EDGE and the current study are worth noting, however. An important consideration of EDGE is the influence of minor allele frequency on the assignment of the estimated heterozygous value. Despite this effect, EDGE did maintain high average power across simulated interaction models and conserved type I error. As such, we thought it possible that EDGE may be able to detect interaction between SNPs with lower MAF than 5% and did not apply a MAF QC filter. Given the significant interaction for cataract involving rs7787286

(MAF: 0.041), we see EDGE as a potential method for exploring low frequency variants for epistasis. Future work will focus on evaluating EDGE with low frequency variant simulations and applying it to low frequency variants in natural data. An additional limitation is that our study design focused on 2 SNP interactions. This was an important first step, but the complexity of biology may involve epistasis between many more loci, which needs to be considered. Finally, it is important to note that, as with other genetic association studies, SNPs found to be significant in this study may be tagging functional variation and not directly related to the phenotypes we evaluated.

Future research will involve investigating EDGE's power to detect simulated interactions in the presence of main effect(s). This scenario is likely reflective of biological cases in which pure epistasis is not at play. Questions also remain regarding the consistency of EDGE-derived heterozygous assignment across different datasets and populations. We had the opportunity to explore this for the SNPs in the epistasis models we selected for replication, which showed that most of these SNPs had similar alpha assignments in the different datasets, but this will be explored for genome-wide SNPs in future work. Applications of EDGE to gene-environment interactions are in development, including key areas that need to be explored in the context of the exposome, including: 1) how the alpha value may vary across different environmental exposure contexts and 2) the impact of exposure variable type (i.e., categorical versus continuous) on the ability to detect EDGE encoded SNP x environment interactions. Finally, future work will involve extending EDGE to machine learning interaction methodologies, which will involve rigorous assessment and comparison to methods including multifactor dimensionality reduction (MDR) [42], neural networks [43,44], random forests [45], learning classifier systems [46], and Bayesian Networks [47].

We demonstrated the utility of our novel encoding method at flexibly assigning a genetic model to individual loci, identifying interactions between SNPs with diverse genetic models, and uncovering examples of nonadditive allelic architecture associated with complex disease. With the development of EDGE, we offer a method that determines an individualized SNP encoding. Applying EDGE to SNP data for epistasis research will allow discovery of SNP-SNP models that have been undetected because their genetic model is nonadditive.

## Methods

### Simulated datasets

To assess the ability of EDGE to accurately assign heterozygous genotype values and identify SNP-SNP interactions across different genetic models with high power and low type I error, we developed the Biallelic Model Simulator (available at https://www.hall-lab.org/). This script generates two independent, biallelic SNPs in Hardy-Weinberg equilibrium according to given minor allele frequencies (for a further description of the simulation method, see S1 Text).

To evaluate multiple scenarios for epistasis to occur between a pair of SNPs [48], we simulated interactions with no main effect between SNP1 and SNP2 across pairwise combinations of underlying genetic models, including all two-way combinations between the following models: additive (ADD), dominant (DOM), recessive (REC), sub-additive (SUB), super-additive (SUP), and heterozygous (HET), referred to as "pairwise SNP-SNP" interaction models. Here, heterozygous genotypes are simulated to have half (ADD), the same (DOM), zero (REC), 25% (SUB), and 75% (SUP) the risk of homozygous alternate genotypes. Heterozygous genotypes for HET SNPs were simulated to have full risk relative to both homozygous genotypes. Additionally, we evaluated simulated interactions using genotype-base interaction models that include penetrance functions (e.g., XOR, Hyperbolic) and scenarios in which only one of the 9 interaction penetrance cells confers risk while the other 8 demonstrate no risk (e.g.,

**Table 3. Genotype-based simulated models.**

|  | HR-HR | HR-HET | HR-HA | HET-HET | HET-HA | HA-HA | XOR | Hyp | RHyp |
|---|---|---|---|---|---|---|---|---|---|
| AABB | 1 | 0 | 0 | 0 | 0 | 0 | 1 | 0 | 1 |
| AABb | 0 | 1 | 0 | 0 | 0 | 0 | 0 | 0.5 | 0.5 |
| AAbb | 0 | 0 | 1 | 0 | 0 | 0 | 1 | 1 | 0 |
| AaBB | 0 | 0 | 0 | 0 | 0 | 0 | 0 | 0.5 | 0.5 |
| AaBb | 0 | 0 | 0 | 1 | 0 | 0 | 1 | 0.5 | 0.5 |
| Aabb | 0 | 0 | 0 | 0 | 1 | 0 | 0 | 0.5 | 0.5 |
| aaBB | 0 | 0 | 0 | 0 | 0 | 0 | 1 | 1 | 0 |
| aaBb | 0 | 0 | 0 | 0 | 0 | 0 | 0 | 0.5 | 0.5 |
| aabb | 0 | 0 | 0 | 0 | 0 | 1 | 1 | 0 | 1 |

HR: Homozygous Referent

HET: Heterozygous

HA: Homozygous Alternate

XOR: XOR Model

Hyp: Hyperbolic Model

RHyp: Hyperbolic Model

Homozygous Referent-Homozygous Referent–HR-HR) (Table 3 lists all genotype-based models). To assess the impact of the aforementioned parameters, we performed a parameter test by simulating all models with every combination of the following: MAF: 0.05, 0.1, 0.3, 0.5; sample size: 2,000, 10,000, and 20,000; case-control ratio: 1:1 (balanced), 1:3, and 3:1 (unbalanced), and baseline risk.

To test for type I error, 1,000 null signal datasets were simulated for each combination of the parameters described above. In addition, to assess inflation driven by the encoding when models demonstrate main effects without interaction signal, we simulated datasets under two scenarios. The first involved simulating two SNPs, both exhibiting a main effect but having no interaction effect ("Two-SNP Main Effect"). The second involved simulating two SNPs in which only one SNP exhibited a main effect and with no interaction effect ("One-SNP Main Effect"). For the Two-SNP Main Effect models, underlying biological model combinations between REC, SUB, ADD, SUP, DOM, and HET were simulated. For the One-SNP Main Effect models, we simulated datasets allowing for only one SNP to exhibit the above biological models.

## eMERGE datasets

Genome-wide genotyping was performed on approximately 55,000 samples (397 of Asian ancestry, 11,109 of African ancestry, 40,243 of European ancestry, 108 of Native American ancestry, and 3,167 of unknown ancestry) across the eMERGE II study sites at the Broad Institute and at the Center for Inherited Disease Research (CIDR) using the Illumina 660W-Quad or 1M-Duo BeadChips. In eMERGE, genetic data is imputed to 1000 genomes reference panel (March 2012 release). Imputation was performed on datasets separated by site and platform using IMPUTE2 on the phased genotyped data (SHAPEIT2 was used for phasing). The following sites were included for analysis: Geisinger (AMD, Glaucoma, RH, T2D), Group Health (AMD, Cataract, Glaucoma, RH, T2D), Marshfield (AMD, Cataract, Glaucoma, RH, T2D), Mayo (AMD, Cataract, Glaucoma, RH, T2D), Mount Sinai (RH, T2D), Northwestern (AMD, Glaucoma, RH, T2D), Vanderbilt (AMD, Cataract, Glaucoma, RH, T2D). Data were cleaned using the eMERGE QC pipeline developed by the eMERGE Genomics Working Group [28].

This process includes evaluation of sample and marker call rate, sex mismatch, duplicate and HapMap concordance, batch effects, Hardy-Weinberg equilibrium, sample relatedness, and population stratification. For all phenotypes, QC filters with 99% marker and sample call rates were applied. SNPs were LD pruned using an r-square threshold of 0.7.

For the T2D phenotype, there were 20,341 total samples (7,101 cases and 13,240 controls, 53% female), and following QC and LD pruning, 358,569 SNPs were available for analysis. The age-related cataract analysis included 6,815 total samples (5,104 cases and 1,711 controls, 56% females) and 333,898 post-QC SNPs. For glaucoma, 5,090 total samples (961 cases and 4,129 controls, 56% female) and 348,329 post-QC SNPs were considered. AMD included 13,153 total samples (2,167 cases and 10,986 controls, 54% female) and 311,161 post-QC SNPs. Finally, for resistant hypertension, 3,706 total samples (2,830 cases and 876 controls, 58% female) and 368,528 post-QC SNPs were included for analysis.

## Replication datasets

Candidate replication SNP-SNP interaction analyses were performed on data from the UK Biobank (UKB) [30]. The UK Biobank contains genetic and phenotypic data on approximately 500,000 individuals. For genotypic quality, we removed 35,785 poor quality samples determined to be outliers for heterozygosity and/or missing rate as well as individuals found to be related based on a Pi-hat of 0.25. Originally genotyped to 96 million variants, we excluded variants that had an imputation info score less than 0.3 and pruned the data at an r-square threshold of 0.7. Additionally, we retained only individuals of European ancestry as inferred by UKB due to the skew in sample size of this ancestry group compared to others in UKB.

The PheWAS R package [49] was used to determine ICD-9 and ICD-10 based phecode phenotypes. The following phecodes were used in analysis: 362.2 degeneration of macula and posterior pole of retina (AMD), 366 cataract, and 401.1 essential hypertension, and 250.2 type 2 diabetes. Note that data was not available for resistant hypertension; however, hypertension was considered in UKB to explore potential similarities of epistatic signal across the two related phenotypes. A subject was considered a case for the phecode if they had one instance of an ICD code mapping to that phecode and a control if they had 0 instances of an ICD code mapping to that phecode and did not meet the exclusion criteria as defined in the PheWAS R package. The resulting datasets included 1,992 with AMD (286,169 controls), 16,574 with age-related cataract (264,905 controls), 16,621 with T2D (276,718 controls), and 70,089 with hypertension (219,326 controls) in UKB. If the exact SNP-SNP interaction did not replicate in UKB (as was the case for T2D and age-related cataract), we performed a region-based SNP-SNP interaction analysis, whereby interactions were considered between SNPs within a 50kb window upstream and downstream of the original SNP-SNP interaction models to allow for differences in LD structure and MAF across the datasets. To control for multiple tests in the region-based replication analyses, we applied a Bonferroni adjustment for the number of SNP-SNP models.

## Statistical analyses

For all simulated and eMERGE datasets, regression modelling was performed using PLATO software [31], which employs EDGE, additive, dominant, recessive, and dominant encodings with user specification. *Multi-encoding GWAS*: In the eMERGE dataset, we performed four GWAS for each phenotype: each GWAS employing one of the traditional encodings (i.e., additive-encoded GWAS, dominant-encoded GWAS, recessive-encoded GWAS, and codominant-encoded GWAS) using logistic regression. We created pairwise scatterplots of the -log10 p-values from the results of each of the four traditional genetic encodings, additive, codominant,

dominant, and recessive, to better visualize how concordant the p-values were across these encodings for the AMD and T2D phenotypes.

*Interaction analysis*: For both simulated and natural data, to determine the significance of a SNP-SNP interaction model above and beyond the main effects of both SNPs combined, we performed a likelihood ratio test (LRT) between the full ($Y = \beta_0 + \beta_1 SNP1 + \beta_2 SNP2 + \beta_3 SNP1 \times SNP2$) and reduced ($Y = \beta_0 + \beta_1 SNP1 + \beta_2 SNP2$) models and derived an LRT p-value using each of the five encodings separately: additive, dominant, recessive, codominant, and EDGE. Select covariates were included in the regression models as well, as described below. In order to reduce computational and multiple testing burden in the eMERGE interaction analysis, a main effect filter was applied. So as to ensure that SNPs with diverse underlying genetic models were selected, the top 50 SNPs from each encoding were selected for subsequent interaction analysis. We chose this approach over a specified significance threshold so as not to bias any one encoding type if one encoding yielded more results from each multi-encoding GWAS. Pairwise combinations of interactions were tested between SNPs within the union of the top 50 most significant SNPs, and SNP-SNP LRT tests were performed with additive, dominant, recessive, codominant, and EDGE encodings separately.

*Covariate adjustment*. In eMERGE, for all phenotypes, regression models were tested while adjusting for sex, decade of birth, eMERGE site, genotyping platform, and BMI. To adjust for population stratification, PCA analysis was performed using Eigenstrat [50] and we included as covariates the first 6 (T2D and glaucoma), 3 (cataract and AMD), and 10 (resistant hypertension) principal components (number of principal components included as covariates varied across phenotypes due to differences in the amount of variance explained by principal components for each individual phenotype).

*Testing the impact of parameters on power*. To test the impact of allele frequency, sample size, penetrance, and case-control ratio on power, we performed a parameter test. All 29 interacting models were simulated with comprehensive combinations of each of the four parameters. After compiling all power results from simulated datasets of our parameter sweep, we performed ANOVA tests of the power by model for each encoding to identify effects of each parameter on power for every genetic model type.

## Replication analyses

For replication analyses in UKB, we extracted SNP-SNP interaction models found to be significant in eMERGE for AMD, T2D, age-related cataract, and hypertension phenotypes. PLATO software was used to run logistic regression for each of these models, again considering each of the additive, codominant, dominant, recessive, and EDGE encodings separately. Each model was adjusted for age, sex, BMI, and principal components (first 10 UKB-generated PCs).

## Supporting information

**S1 Text. Simulation Description.**
(DOCX)

**S1 Fig. Impact of MAF on alpha value.**
(DOCX)

**S2 Fig. Results of the multi-encoding GWAS for age-related cataract, glaucoma, and resistance hypertension.**
(DOCX)

**S1 Table. Results from T2D multi-encoding with additive (A), dominant (B), recessive (C), codominant (D), as well as the alpha value for each SNP (categ_weight) (E).** Var1_ID

(SNP's rs#), Var1_Pos (chromosome:base pair), Var1_MAF (SNP's minor allele frequency), Num_nonMissing (sample size), Num_Cases (number of cases), N_Iter (number of iterations), Converged (model convergence; yes:1, no:0), Raw_LRT_pval (unadjusted likelihood ratio test p-value), Var1_Pval (SNP p-value), Var1_beta (SNP beta value), Overall_LRT_Pval (Likelihood ratio test p-value), Overall_Pval_adj_Bonferroni (Bonferroni adjusted SNP p-value), Overall_Pval_adj_FDR (FDR adjusted SNP p-value).
(XLSX)

**S2 Table. Results from AMD multi-encoding with additive (A), dominant (B), recessive (C), codominant (D), as well as the alpha value for each SNP (categ_weight) (E).** Var1_ID (SNP's rs#), Var1_Pos (chromosome:base pair), Var1_MAF (SNP's minor allele frequency), Num_nonMissing (sample size), Num_Cases (number of cases), N_Iter (number of iterations), Converged (model convergence; yes:1, no:0), Raw_LRT_pval (unadjusted likelihood ratio test p-value), Var1_Pval (SNP p-value), Var1_beta (SNP beta value), Overall_LRT_Pval (Likelihood ratio test p-value), Overall_Pval_adj_Bonferroni (Bonferroni adjusted SNP p-value), Overall_Pval_adj_FDR (FDR adjusted SNP p-value).
(XLSX)

**S3 Table. Results from cataract multi-encoding with additive (A), dominant (B), recessive (C), codominant (D), as well as the alpha value for each SNP (categ_weight) (E).** Var1_ID (SNP's rs#), Var1_Pos (chromosome:base pair), Var1_MAF (SNP's minor allele frequency), Num_nonMissing (sample size), Num_Cases (number of cases), N_Iter (number of iterations), Converged (model convergence; yes:1, no:0), Raw_LRT_pval (unadjusted likelihood ratio test p-value), Var1_Pval (SNP p-value), Var1_beta (SNP beta value), Overall_LRT_Pval (Likelihood ratio test p-value), Overall_Pval_adj_Bonferroni (Bonferroni adjusted SNP p-value), Overall_Pval_adj_FDR (FDR adjusted SNP p-value).
(XLSX)

**S4 Table. Results from resistant hypertension multi-encoding with additive (A), dominant (B), recessive (C), codominant (D), as well as the alpha value for each SNP (categ_weight) (E).** Var1_ID (SNP's rs#), Var1_Pos (chromosome:base pair), Var1_MAF (SNP's minor allele frequency), Num_nonMissing (sample size), Num_Cases (number of cases), N_Iter (number of iterations), Converged (model convergence; yes:1, no:0), Raw_LRT_pval (unadjusted likelihood ratio test p-value), Var1_Pval (SNP p-value), Var1_beta (SNP beta value), Overall_LRT_Pval (Likelihood ratio test p-value), Overall_Pval_adj_Bonferroni (Bonferroni adjusted SNP p-value), Overall_Pval_adj_FDR (FDR adjusted SNP p-value).
(XLSX)

**S5 Table. Results from glaucoma multi-encoding with additive (A), dominant (B), recessive (C), codominant (D), as well as the alpha value for each SNP (categ_weight) (E).** Var1_ID (SNP's rs#), Var1_Pos (chromosome:base pair), Var1_MAF (SNP's minor allele frequency), Num_nonMissing (sample size), Num_Cases (number of cases), N_Iter (number of iterations), Converged (model convergence; yes:1, no:0), Raw_LRT_pval (unadjusted likelihood ratio test p-value), Var1_Pval (SNP p-value), Var1_beta (SNP beta value), Overall_LRT_Pval (Likelihood ratio test p-value), Overall_Pval_adj_Bonferroni (Bonferroni adjusted SNP p-value), Overall_Pval_adj_FDR (FDR adjusted SNP p-value).
(XLSX)

**S6 Table. Results from SNP-SNP interactions for T2D in eMERGE with additive (A), dominant (B), recessive (C), codominant (D), and EDGE (E) encodings.** Var1_ID (SNP 1 rs#), Var1_Pos (SNP 1 chromosome:base pair), Var1_MAF (SNP 1 minor allele frequency),

Var1_ID (SNP's rs#), Var2_Pos (SNP 2 chromosome:base pair), Var2_MAF (SNP 2 minor allele frequency), Num_nonMissing (sample size), Num_Cases (number of cases), N_Iter (number of iterations), Converged (model convergence; yes:1, no:0), Raw_LRT_pval (unadjusted likelihood ratio test p-value), Red_Var1_Pval (SNP 1 p-value in reduced model), Red_Var1_beta (SNP 1 beta value in reduced model), Red_Var1_SE (SNP 1 standard error in reduced model), Red_Var2_Pval (SNP 2 p-value in reduced model), Red_Var2_beta (SNP 2 beta value in reduced model), Red_Var2_SE (SNP 2 standard error in reduced model), Full_Var1_Pval (SNP 1 p-value in full model), Full_Var1_beta (SNP 1 beta value in full model), Full_Var1_SE (SNP 1 standard error in full model), Full_Var2_Pval (SNP 2 p-value in full model), Full_Var2_beta (SNP 2 beta value in full model), Full_Var2_SE (SNP 2 standard error in full model), Full_Var1_Var2_Pval (p-value of interaction between SNP1 and SNP2 in full model), Full_Var1_Var2_beta (beta value of interaction between SNP1 and SNP2 in full model), Full_Var1_Var2_SE (standard error of interaction between SNP1 and SNP2 in full model), Red_model_Pval (reduced model p-value), Full_model_Pval (full model p-value), Overall_LRT_Pval (Likelihood ratio test p-value), Overall_Pval_adj_Bonferroni (Bonferroni adjusted LRT p-value), Overall_Pval_adj_FDR (FDR adjusted LRT p-value). (XLSX)

**S7 Table. Results from SNP-SNP interactions for cataract in eMERGE with additive (A), dominant (B), recessive (C), codominant (D), and EDGE (E) encodings.** Var1_ID (SNP 1 rs#), Var1_Pos (SNP 1 chromosome:base pair), Var1_MAF (SNP 1 minor allele frequency), Var1_ID (SNP's rs#), Var2_Pos (SNP 2 chromosome:base pair), Var2_MAF (SNP 2 minor allele frequency), Num_nonMissing (sample size), Num_Cases (number of cases), N_Iter (number of iterations), Converged (model convergence; yes:1, no:0), Raw_LRT_pval (unadjusted likelihood ratio test p-value), Red_Var1_Pval (SNP 1 p-value in reduced model), Red_Var1_beta (SNP 1 beta value in reduced model), Red_Var1_SE (SNP 1 standard error in reduced model), Red_Var2_Pval (SNP 2 p-value in reduced model), Red_Var2_beta (SNP 2 beta value in reduced model), Red_Var2_SE (SNP 2 standard error in reduced model), Full_Var1_Pval (SNP 1 p-value in full model), Full_Var1_beta (SNP 1 beta value in full model), Full_Var1_SE (SNP 1 standard error in full model), Full_Var2_Pval (SNP 2 p-value in full model), Full_Var2_beta (SNP 2 beta value in full model), Full_Var2_SE (SNP 2 standard error in full model), Full_Var1_Var2_Pval (p-value of interaction between SNP1 and SNP2 in full model), Full_Var1_Var2_beta (beta value of interaction between SNP1 and SNP2 in full model), Full_Var1_Var2_SE (standard error of interaction between SNP1 and SNP2 in full model), Red_model_Pval (reduced model p-value), Full_model_Pval (full model p-value), Overall_LRT_Pval (Likelihood ratio test p-value), Overall_Pval_adj_Bonferroni (Bonferroni adjusted LRT p-value), Overall_Pval_adj_FDR (FDR adjusted LRT p-value). (XLSX)

**S8 Table. Results from SNP-SNP interactions for cataract in UKB with additive (A), dominant (B), recessive (C), codominant (D), and EDGE (E) encodings.** Var1_ID (SNP 1 rs#), Var1_Pos (SNP 1 chromosome:base pair), Var1_MAF (SNP 1 minor allele frequency), Var1_ID (SNP's rs#), Var2_Pos (SNP 2 chromosome:base pair), Var2_MAF (SNP 2 minor allele frequency), Num_nonMissing (sample size), Num_Cases (number of cases), N_Iter (number of iterations), Converged (model convergence; yes:1, no:0), Raw_LRT_pval (unadjusted likelihood ratio test p-value), Red_Var1_Pval (SNP 1 p-value in reduced model), Red_Var1_beta (SNP 1 beta value in reduced model), Red_Var1_SE (SNP 1 standard error in reduced model), Red_Var2_Pval (SNP 2 p-value in reduced model), Red_Var2_beta (SNP 2 beta value in reduced model), Red_Var2_SE (SNP 2 standard error in reduced model), Full_Var1_Pval (SNP 1 p-value in full model), Full_Var1_beta (SNP 1 beta value in full model),

Full_Var1_SE (SNP 1 standard error in full model), Full_Var2_Pval (SNP 2 p-value in full model), Full_Var2_beta (SNP 2 beta value in full model), Full_Var2_SE (SNP 2 standard error in full model), Full_Var1_Var2_Pval (p-value of interaction between SNP1 and SNP2 in full model), Full_Var1_Var2_beta (beta value of interaction between SNP1 and SNP2 in full model), Full_Var1_Var2_SE (standard error of interaction between SNP1 and SNP2 in full model), Red_model_Pval (reduced model p-value), Full_model_Pval (full model p-value), Overall_LRT_Pval (Likelihood ratio test p-value), Overall_Pval_adj_Bonferroni (Bonferroni adjusted LRT p-value), Overall_Pval_adj_FDR (FDR adjusted LRT p-value). (XLSX)

**S9 Table. Results from SNP-SNP interactions for resistant hypertension in eMERGE with additive (A), dominant (B), recessive (C), codominant (D), and EDGE (E) encodings.** Var1_ID (SNP 1 rs#), Var1_Pos (SNP 1 chromosome:base pair), Var1_MAF (SNP 1 minor allele frequency), Var1_ID (SNP's rs#), Var2_Pos (SNP 2 chromosome:base pair), Var2_MAF (SNP 2 minor allele frequency), Num_nonMissing (sample size), Num_Cases (number of cases), N_Iter (number of iterations), Converged (model convergence; yes:1, no:0), Raw_-LRT_pval (unadjusted likelihood ratio test p-value), Red_Var1_Pval (SNP 1 p-value in reduced model), Red_Var1_beta (SNP 1 beta value in reduced model), Red_Var1_SE (SNP 1 standard error in reduced model), Red_Var2_Pval (SNP 2 p-value in reduced model), Red_-Var2_beta (SNP 2 beta value in reduced model), Red_Var2_SE (SNP 2 standard error in reduced model), Full_Var1_Pval (SNP 1 p-value in full model), Full_Var1_beta (SNP 1 beta value in full model), Full_Var1_SE (SNP 1 standard error in full model), Full_Var2_Pval (SNP 2 p-value in full model), Full_Var2_beta (SNP 2 beta value in full model), Full_Var2_SE (SNP 2 standard error in full model), Full_Var1_Var2_Pval (p-value of interaction between SNP1 and SNP2 in full model), Full_Var1_Var2_beta (beta value of interaction between SNP1 and SNP2 in full model), Full_Var1_Var2_SE (standard error of interaction between SNP1 and SNP2 in full model), Red_model_Pval (reduced model p-value), Full_model_Pval (full model p-value), Overall_LRT_Pval (Likelihood ratio test p-value), Overall_Pval_adj_Bonferroni (Bonferroni adjusted LRT p-value), Overall_Pval_adj_FDR (FDR adjusted LRT p-value). (XLSX)

**S10 Table. Results from SNP-SNP interactions for hypertension in UKB with additive (A), dominant (B), recessive (C), codominant (D), and EDGE (E) encodings.** Var1_ID (SNP 1 rs#), Var1_Pos (SNP 1 chromosome:base pair), Var1_MAF (SNP 1 minor allele frequency), Var1_ID (SNP's rs#), Var2_Pos (SNP 2 chromosome:base pair), Var2_MAF (SNP 2 minor allele frequency), Num_nonMissing (sample size), Num_Cases (number of cases), N_Iter (number of iterations), Converged (model convergence; yes:1, no:0), Raw_LRT_pval (unadjusted likelihood ratio test p-value), Red_Var1_Pval (SNP 1 p-value in reduced model), Red_-Var1_beta (SNP 1 beta value in reduced model), Red_Var1_SE (SNP 1 standard error in reduced model), Red_Var2_Pval (SNP 2 p-value in reduced model), Red_Var2_beta (SNP 2 beta value in reduced model), Red_Var2_SE (SNP 2 standard error in reduced model), Full_-Var1_Pval (SNP 1 p-value in full model), Full_Var1_beta (SNP 1 beta value in full model), Full_Var1_SE (SNP 1 standard error in full model), Full_Var2_Pval (SNP 2 p-value in full model), Full_Var2_beta (SNP 2 beta value in full model), Full_Var2_SE (SNP 2 standard error in full model), Full_Var1_Var2_Pval (p-value of interaction between SNP1 and SNP2 in full model), Full_Var1_Var2_beta (beta value of interaction between SNP1 and SNP2 in full model), Full_Var1_Var2_SE (standard error of interaction between SNP1 and SNP2 in full model), Red_model_Pval (reduced model p-value), Full_model_Pval (full model p-value), Overall_LRT_Pval (Likelihood ratio test p-value), Overall_Pval_adj_Bonferroni (Bonferroni

adjusted LRT p-value), Overall_Pval_adj_FDR (FDR adjusted LRT p-value).
(XLSX)

**S11 Table. Results from SNP-SNP interactions for AMD in eMERGE with additive (A), dominant (B), recessive (C), codominant (D), and EDGE (E) encodings.** Var1_ID (SNP 1 rs#), Var1_Pos (SNP 1 chromosome:base pair), Var1_MAF (SNP 1 minor allele frequency), Var1_ID (SNP's rs#), Var2_Pos (SNP 2 chromosome:base pair), Var2_MAF (SNP 2 minor allele frequency), Num_nonMissing (sample size), Num_Cases (number of cases), N_Iter (number of iterations), Converged (model convergence; yes:1, no:0), Raw_LRT_pval (unadjusted likelihood ratio test p-value), Red_Var1_Pval (SNP 1 p-value in reduced model), Red_Var1_beta (SNP 1 beta value in reduced model), Red_Var1_SE (SNP 1 standard error in reduced model), Red_Var2_Pval (SNP 2 p-value in reduced model), Red_Var2_beta (SNP 2 beta value in reduced model), Red_Var2_SE (SNP 2 standard error in reduced model), Full_Var1_Pval (SNP 1 p-value in full model), Full_Var1_beta (SNP 1 beta value in full model), Full_Var1_SE (SNP 1 standard error in full model), Full_Var2_Pval (SNP 2 p-value in full model), Full_Var2_beta (SNP 2 beta value in full model), Full_Var2_SE (SNP 2 standard error in full model), Full_Var1_Var2_Pval (p-value of interaction between SNP1 and SNP2 in full model), Full_Var1_Var2_beta (beta value of interaction between SNP1 and SNP2 in full model), Full_Var1_Var2_SE (standard error of interaction between SNP1 and SNP2 in full model), Red_model_Pval (reduced model p-value), Full_model_Pval (full model p-value), Overall_LRT_Pval (Likelihood ratio test p-value), Overall_Pval_adj_Bonferroni (Bonferroni adjusted LRT p-value), Overall_Pval_adj_FDR (FDR adjusted LRT p-value).
(XLSX)

**S12 Table. Results from SNP-SNP interactions for AMD in UKB with additive (A), dominant (B), recessive (C), codominant (D), and EDGE (E) encodings.** Var1_ID (SNP 1 rs#), Var1_Pos (SNP 1 chromosome:base pair), Var1_MAF (SNP 1 minor allele frequency), Var1_ID (SNP's rs#), Var2_Pos (SNP 2 chromosome:base pair), Var2_MAF (SNP 2 minor allele frequency), Num_nonMissing (sample size), Num_Cases (number of cases), N_Iter (number of iterations), Converged (model convergence; yes:1, no:0), Raw_LRT_pval (unadjusted likelihood ratio test p-value), Red_Var1_Pval (SNP 1 p-value in reduced model), Red_Var1_beta (SNP 1 beta value in reduced model), Red_Var1_SE (SNP 1 standard error in reduced model), Red_Var2_Pval (SNP 2 p-value in reduced model), Red_Var2_beta (SNP 2 beta value in reduced model), Red_Var2_SE (SNP 2 standard error in reduced model), Full_Var1_Pval (SNP 1 p-value in full model), Full_Var1_beta (SNP 1 beta value in full model), Full_Var1_SE (SNP 1 standard error in full model), Full_Var2_Pval (SNP 2 p-value in full model), Full_Var2_beta (SNP 2 beta value in full model), Full_Var2_SE (SNP 2 standard error in full model), Full_Var1_Var2_Pval (p-value of interaction between SNP1 and SNP2 in full model), Full_Var1_Var2_beta (beta value of interaction between SNP1 and SNP2 in full model), Full_Var1_Var2_SE (standard error of interaction between SNP1 and SNP2 in full model), Red_model_Pval (reduced model p-value), Full_model_Pval (full model p-value), Overall_LRT_Pval (Likelihood ratio test p-value), Overall_Pval_adj_Bonferroni (Bonferroni adjusted LRT p-value), Overall_Pval_adj_FDR (FDR adjusted LRT p-value).
(XLSX)

## Author Contributions

**Conceptualization:** Molly A. Hall, John Wallace, Shefali S. Verma, Bertram Müller-Myhsok, Sarah A. Pendergrass, Kristel Van Steen, Jason H. Moore, Marylyn D. Ritchie.

**Data curation:** Molly A. Hall, John Wallace, Yuki Bradford, Shefali S. Verma.

**Formal analysis:** Molly A. Hall, John Wallace, Anastasia M. Lucas, Yuki Bradford, John McGuigan, Beibei Jiang.

**Funding acquisition:** Jason H. Moore, Marylyn D. Ritchie.

**Investigation:** Molly A. Hall, John Wallace, Anastasia M. Lucas.

**Methodology:** Molly A. Hall, John Wallace, Shefali S. Verma, Bertram Müller-Myhsok, Kristel Van Steen, Marylyn D. Ritchie.

**Project administration:** Jason H. Moore, Marylyn D. Ritchie.

**Resources:** Murray Brilliant, Patrick Sleiman, Hakon Hakonarson, John B. Harley, Krzysztof Kiryluk, Jason H. Moore, Marylyn D. Ritchie.

**Software:** John Wallace, Marylyn D. Ritchie.

**Supervision:** Kristel Van Steen, Jason H. Moore, Marylyn D. Ritchie.

**Validation:** Molly A. Hall, Anastasia M. Lucas, John McGuigan.

**Visualization:** John Wallace, Anastasia M. Lucas.

**Writing – original draft:** Molly A. Hall, Anastasia M. Lucas, Kristin Passero, Jiayan Zhou.

**Writing – review & editing:** Molly A. Hall, Anastasia M. Lucas, Shefali S. Verma, Bertram Müller-Myhsok, Kristin Passero, Jiayan Zhou, Sarah A. Pendergrass, Yanfei Zhang, Peggy Peissig, Murray Brilliant, Patrick Sleiman, Hakon Hakonarson, John B. Harley, Krzysztof Kiryluk, Kristel Van Steen, Jason H. Moore, Marylyn D. Ritchie.

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
