## [Decision Letter · Decision Letter 0]

14 Jul 2020

Dear Dr Hall,

Thank you very much for submitting your Research Article entitled 'Novel encoding method EDGE enhances ability to identify genetic interactions' to PLOS Genetics. Your manuscript was fully evaluated at the editorial level and by independent peer reviewers. The reviewers appreciated the attention to an important problem, but raised some substantial concerns about the current manuscript. Based on the reviews, we will not be able to accept this version of the manuscript, but we would be willing to review again a much-revised version. We cannot, of course, promise publication at that time.

If you decide to revise the manuscript for further consideration at PLOS Genetics, please aim to resubmit within the next 60 days, unless it will take extra time to address the concerns of the reviewers, in which case we would appreciate an expected resubmission date by email to plosgenetics@plos.org.

[LINK]

We are sorry that we cannot be more positive about your manuscript at this stage. Please do not hesitate to contact us if you have any concerns or questions.

Yours sincerely,

Heather J Cordell

Associate Editor

PLOS Genetics

Hua Tang

Section Editor: Natural Variation

PLOS Genetics

Reviewer's Responses to Questions

**Comments to the Authors:**

Reviewer #1: Hall et al. present a new "elastic data-driven genetic encoding" (EDGE) method to enhance identification of genetic associations and epistatic interactions. Instead of being restricted by various genetic model parameters of the heterozygous genotype, EDGE concerns a single parameter alpha as the ratio of the linear regression coefficients of the heterozygous and the homozygous alternate genotypes, and thus is elastic and data-driven. Based on a set of simulation and real data analyses, the authors conclude that the EDGE method is robust and can be used as a flexible alternative for testing pair-wise epistatic interactions among common variants.

While the idea is intriguing, the manuscript is not easy to read and the work seems incomplete with a few key questions can be asked. What concerns me most is that none of the identified epistatic interactions are tested for replication in an independent data. I hope the following comments are useful to improve the manuscript.

Major points:

• Replication. Statistical epistatic interactions remain a challenging topic for many reasons. One reason is lack of robust examples with strong statistical support and validated biological meanings. Statistical replication is an important first step to prevent reporting false positive findings. Considering just the rs7787286 – rs4695885 interaction in age-related cataract identified only by EDGE, the MAF of rs7787286 is 0.035 in Europeans that is lower than 0.05 used in the simulation scenarios and the QC procedures of eMERGE data. Statistical replication is therefore recommended.

• Low frequent variants. As mentioned in the Discussion section, EDGE currently concerns only frequent variants and the authors recommend caution when working with low frequent variants, i.e. MAF less than 10% if I understand correctly. However, recent theoretical studies suggest that common variants identified in GWAS are often associated with high LD and low probability of being causal. It seems necessary to cover low frequent variants in epistasis studies although this means large samples are required to identify their interactions.

• Simulation. If I understand correctly, the simulation scenarios used concern only one pair of SNPs simulated with predefined parameters, but without concerning the genome-wide settings incurred in real data analyses. I think the impact of this limitation needs to be discussed as well.

• Significance threshold. I am a bit nervous about the ways of deriving significance thresholds. I can understand using EDGE as a main effect filter for interaction tests but can understand why using the top 50 (not 30 or 70 for example) most significant SNPs per model to derive the test set, which appears to assume EDGE alone could pick up epistatic SNPs without bias and could lead to varied size of the test set and subsequently jeopardize Bonferroni adjustment.

Minor points:

• For each pair of epistatic interacting SNPs, please report MAF of each and their LD given the pruning QC at 0.7

• Please add detailed MAF information in the Abstract

• The Introduction could be stronger if updates of recent development in studying epistasis could be provided. I am a bit surprised that only 14 references are cited for this difficult topic

• page 3 line 64: 'appled' to 'applied'

• page 5 lines 106 to 107: what are 'the current state-of-the-art methodologies'? what is the current paradigm?

• page 5 lines 108 to 110: 'Results discussed ... for GxG' – move the statement to Discussion?

• page 6 Table 2: 'Codominant (HA)' to 'Homozygous alternate (HA)'

• page 8 line 163: the dominant scenario includes under- and over-dominance?

• page 8 lines 169 to 172: in Figure 1 why super-additive varied a lot more than sub-additive and additive? why presenting only MAF of 0.3?

• page 9 lines 190 to 197: in Figure 2 the genotype model seems doing ok at the LRT cutoff of 0.05?

• line 221: ';,' to ';'

• line 232 and line 242: '50% (B)' to '50% (C)'?

• lines 246 to 262: what MAF in the NULL simulations?

• lines 272: please provide scales for the signal to noise ratios in Figure 7

• line 294: 'demonstrated similar patterns' – sorry but I can't tell

• line 304: 'uncorrected p-value' – why not using corrected p-value instead in Figures 8 to 10

• line 352: rs117537110 not available in dbSNP, a typo?

• lines 515 to 516: 'it is not expected ... under this scenario' – not sure about this claim, perhaps remove it?

• line 613: please specify if LRT p-values are derived using the genotype encoding or other settings, with consistent degrees of freedom.

Reviewer #2: Please see attached comments.

Reviewer #3: This paper proposes a new and more flexible encoding method named EDGE to assign coefficients to the heterozygous alleles for different SNPs. The authors claim that this new encoding method provides higher power in detecting SNP-SNP interactions than existing methods which make rigid assumptions about the underlying interaction models. Experiments using simulation and real data are shown to illustrate the claim.

In GWAS, there are at least about 51 different SNP-SNP interaction models [Li and Reich 2000]. Enumerating all these models is indeed a difficult task. The motivation of this paper is very nice. There are, however, some issues that the authors may wish to address:

1. It is unclear to me how the authors infer the right model for each interacting SNP-SNP pair. If they just use Eq. (1) and Eq. (2), then probably the authors will decide the model type based on the value of \\alpha. In table 2, the authors have shown some typical examples of additive model, dominant model, recessive model, etc. I notice that the \\alpha values in these models are separated at least by 0.25. It is unclear if the authors will choose the model type based on this interval or based on the actual \\alpha value. Also, it is unclear if the authors will carry out any model validation.

2. The power analysis and the comparison with existing methods need to have more elaboration. For example, how much of the power gain is due to the flexibility of EDGE in model determination (based on \\alpha value estimation) and how much is due to the detection method itself? This could be demonstrated in the simulation experiments by generating simulation data with a well-defined interaction model and then compare the performance of EDGE and that of existing methods which assume exactly the same interaction model as in the simulation.

3. There are already some widely-used SNP-SNP interaction detection methods, such as PLINK, MDR, BEAM, and BOOST. The authors should show comparison results with respect to these methods to showcase the advantages of their new method EDGE.

Reference:

Li W, Reich J: A complete enumeration and classification of two-locus diseasemodels. Hum Hered 2000, 50:334–349.

**Have all data underlying the figures and results presented in the manuscript been provided?**

Reviewer #1: Yes

Reviewer #2: **No: **Data is from eMERGE and is not publicly available.

Reviewer #3: Yes

PLOS authors have the option to publish the peer review history of their article (what does this mean?). If published, this will include your full peer review and any attached files.

Reviewer #1: No

Reviewer #2: **Yes: **Brandon J. Coombes

Reviewer #3: No

---

## [Decision Letter · Decision Letter 1]

6 Apr 2021

Dear Dr Hall,

We are pleased to inform you that your manuscript entitled "Novel EDGE encoding method enhances ability to identify genetic interactions" has been editorially accepted for publication in PLOS Genetics. Congratulations!

You may also choose to make minor changes to address the one remaining reviewer comment, if you wish.

Yours sincerely,

Heather J Cordell

Associate Editor

PLOS Genetics

Hua Tang

Section Editor: Natural Variation

PLOS Genetics

Comments from the reviewers (if applicable):

Reviewer's Responses to Questions

**Comments to the Authors:**

Reviewer #1: Thanks for addressing my comments satisfactorily. I have no further comments.

Reviewer #2: The authors have sufficiently addressed my concerns. Thanks

Reviewer #3: The manuscript is substantially revised and re-written. Some of my confusing parts have been clarified.

I have the following comments regarding the responses from the authors:

1. I understand better now about the model. Thanks for the clarification.

2. The second question was due to misunderstanding of the model description in the first version. Now it is resolved.

3. I understand that the authors like to emphasize the importance of new encoding method in SNP-SNP detection and therefore want to only focus on the comparison of different encoding methods in this paper. It would be nice if the authors could discuss more about how much difference the new encoding method will make when compared with many existing SNP-SNP detection methods. If they are doing such comparisons in their on-going work, they may point this out, as they have shown in the response letter.

**Have all data underlying the figures and results presented in the manuscript been provided?**

Reviewer #1: Yes

Reviewer #2: Yes

Reviewer #3: Yes

PLOS authors have the option to publish the peer review history of their article (what does this mean?). If published, this will include your full peer review and any attached files.

Reviewer #1: **Yes: **Wen-Hua Wei

Reviewer #2: **Yes: **Brandon Coombes

Reviewer #3: No

**Data Deposition**

http://datadryad.org/submit?journalID=pgenetics&manu=PGENETICS-D-20-00725R1

**Press Queries**

---

## [Editor Report · Acceptance letter]

2 Jun 2021

PGENETICS-D-20-00725R1 

Novel EDGE encoding method enhances ability to identify genetic interactions 

Dear Dr Hall, 

We are pleased to inform you that your manuscript entitled "Novel EDGE encoding method enhances ability to identify genetic interactions" has been formally accepted for publication in PLOS Genetics! Your manuscript is now with our production department and you will be notified of the publication date in due course.

With kind regards,

Olena Szabo

PLOS Genetics

On behalf of:
